# Waning and boosting of antibody Fc-effector functions upon SARS-CoV-2 vaccination

X. Tong[1,7], R. P. McNamara[1,7], M. J. Avendaño[2], E. F. Serrano[2], T. García-Salum[2,3,4], C. Pardo-Roa[2,3], H. L. Bertera [1], T. M. Chicz [1], J. Levican[2], E. Poblete[2], E. Salinas[2,3], A. Muñoz[2], A. Riquelme[3,5], G. Alter [1] ✉ & R. A. Medina [2,3,4,6] ✉

Since the emergence of SARS-CoV-2, vaccines targeting COVID-19 have been developed with unprecedented speed and efficiency. CoronaVac, utilising an inactivated form of the COVID-19 virus and the mRNA26 based Pfizer/BNT162b2 vaccines are widely distributed. Beyond the ability of vaccines to induce production of neutralizing antibodies, they might lead to the generation of antibodies attenuating the disease by recruiting cytotoxic and opsonophagocytic functions. However, the Fc-effector functions of vaccine induced antibodies are much less studied than virus neutralization. Here, using systems serology, we follow the longitudinal Fc-effector profiles induced by CoronaVac and BNT162b2 up until five months following the two-dose vaccine regimen. Compared to BNT162b2, CoronaVac responses wane more slowly, albeit the levels remain lower than that of BNT162b2 recipients throughout the entire observation period. However, mRNA vaccine boosting of CoronaVac responses, including response to the Omicron variant, induce significantly higher peak of antibody functional responses with increased humoral breadth. In summary, we show that vaccine platform-induced humoral responses are not limited to virus neutralization but rather utilise antibody dependent effector functions. We demonstrate that this functionality wanes with different kinetics and can be rescued and expanded via boosting with subsequent homologous and heterologous vaccination.

Severe acute respiratory syndrome coronavirus-2 (SARS-CoV-2) is the causative agent of coronavirus disease-2019 (COVID-19). Since it was first identified in late 2019[1-3], 11 COVID-19-specific vaccines, using novel and diverse platforms, have been granted the emergency use listing by WHO globally to provide protection against this highly transmissible pathogen, four of these COVID-19 vaccines are currently approved or authorized in the United States[4]. However, despite the remarkable success of the vaccines in protecting the population from the early emergent SARS-CoV-2 viral strains, the virus has undergone adaptations that have facilitated transmission among humans, with mutations selectively accumulating in the receptor-binding domain (RBD), permitting the virus to also escape vaccine-induced neutralizing antibody responses[5-8]. In fact, several variants of concern (VOC) have arisen throughout the world since the onset of the pandemic, causing subsequent waves of transmission[9-14], most strikingly observed with the emergence of the Omicron VOC that led to remarkable global spread[15].

[1]Ragon Institute of MGH, MIT, and Harvard, Cambridge, MA 02139, USA. [2]Department of Pediatric Infectious Diseases and Immunology, School of Medicine, Pontificia Universidad Católica de Chile, Santiago 8331150, Chile. [3]Advanced Interdisciplinary Rehabilitation Register (AIRR) - COVID-19 Working Group, Faculty of Medicine, Pontificia Universidad Católica de Chile, Santiago 8331150, Chile. [4]Department of Pathology and Laboratory Medicine, School of Medicine, Emory University, Atlanta, GA 30322, USA. [5]Department of Gastroenterology, School of Medicine, Pontificia Universidad Catolica de Chile, Santiago 8331150, Chile. [6]Department of Microbiology, Icahn School of Medicine at Mount Sinai, New York, NY 10029, USA. [7]These authors contributed equally: X. Tong, R. P. McNamara. ✉e-mail: galter@mgh.harvard.edu; rafael.medina@emory.edu

Even though these emerging VOCs can subvert neutralization and spread with ease, severe disease and death have not increased proportionally to the spread of the disease. Instead, in unvaccinated populations, it has been documented that all VOCs, including Omicron, have led to severe disease and death[16–18], arguing that the vaccine-induced non-neutralizing immune responses are key to attenuating disease.

Humoral immunity, including both binding and neutralizing antibody titers, has been tightly linked to protection against COVID-19 in phase 3 vaccine trials[19–22]. However, beyond the ability of antibodies to bind and block infection, binding antibodies also can leverage the innate immune system to capture, kill, and clear viruses or infected cells via their ability to interact with Fc-receptors present on all immune cells[23–25]. These non-neutralizing antibody functions selectively evolve in individuals that survive severe disease[26], and are associated with the therapeutic activity of convalescent plasma therapy[27], with vaccine-mediated protection in the non-human primate model[28], and contribute to the therapeutic activity of the monoclonal antibodies[29]. Moreover, recent data suggest that even vaccines using the same technology (i.e., mRNA) can elicit significantly different functional humoral immune responses[24]. Nonetheless, while antibody titers and neutralization wane significantly across vaccine platforms, it is unclear whether functional immunity wanes concomitantly to titers and/or whether the waned immunity can be boosted efficiently.

As of November 2021, approximately 42% of the world's population has received the initial two doses of COVID-19 vaccines[30]. Among the vaccines that have been deployed globally, the inactivated CoronaVac (Sinovac) vaccine and the mRNA BNT162b2 (Pfizer/BioNTech) vaccine have been two of the most broadly deployed vaccines globally, having been administered to billions of individuals. In phase 3 clinical trials, the CoronaVac vaccine exhibited 84% vaccine efficacy against COVID-19 disease, whereas the BNT162b2 vaccine exhibited 95% vaccine efficacy[30–33]. Both clinical trials were conducted in late 2020 and early 2021 in which period Alpha and Gamma were the circulating VOCs in South America, where the study was conducted. Literature reviews and meta-analyses suggested that full vaccination with CoronaVac or BNT162b2 provided strong protection against the Alpha, Beta, Gamma, and Delta variants with a vaccine effectiveness ranging from 70.9% to 96.0% against severe disease. The two vaccines also provided moderate protection against the Omicron variant. According to data published from their respective clinical trials, the most common adverse effects following COVID-19 vaccination include fatigue, fever, muscle pain, headache, and joint pain. More serious side effects were rarely recorded and reported. The clinical trial of the CoronaVac inactivated vaccine reported frequent injection site pain followed by fever and other mild and self-limiting conditions, whereas the Pfizer-BioNTech mRNA vaccine trial reported that the most common adverse effects were mild to moderate fatigue and headache[34]. Differences in antibody and neutralizing titers across the vaccine platforms have been proposed as critical determinants of different efficacy[35]. However, whether these platforms raise distinct overall functional humoral immune responses, wane at similar rates, and whether CoronaVac immunity can be augmented, potentially in the setting of an mRNA-vaccine boost, remains unclear.

Here we deeply profile the functional humoral immune response induced by CoronaVac and Pfizer/BNT162b2 vaccines. Particularly we analyze how the vaccine-induced functional responses wane with time and characterize the boosting capacity of the BNT162b2 vaccine, which enhances the overall humoral responses of CoronaVac primed vaccinees to levels that are higher than the peak responses seen in the recipients vaccinated with two doses of the CoronaVac or BNT162b2 vaccines. We identify that Fc-receptor binding antibodies wane much faster than binding IgG1. Moreover, while antibody functions are induced against Omicron following BNT162b2 vaccination, these responses wane rapidly over time and are not observed in the setting

of CoronaVac vaccination. Importantly, mRNA boosting of the CoronaVac response yields a striking enhancement of functional humoral immunity across VOCs, including Omicron. These data collectively point to marked vaccine platform-based differences in peak functional humoral protection, as well as their distinct waning profiles. This phenotype can be enhanced and, in several cases, functionally expanded by heterologous boosting.

## Results

### COVID-19 vaccines induce antibodies that wane with time
Beyond their ability to neutralize infection, antibodies can attenuate disease via their ability to bind to virus or virally infected cells and then recruit the innate immune system at the site of infection, via Fc-receptors[24,25,27,28]. As a result, the persistence and breadth of binding of Spike-specific antibodies is a key initial determinant of the non-neutralizing capacity of vaccines that protect against COVID-19. Thus, we employed a systems serology approach (Supplementary Figs. 1, 2) to deeply profile the peak immunogenicity and decay profiles of SARS-CoV-2 and VOC-specific antibodies following CoronaVac and Pfizer/BNT162b2 vaccination (Fig. 1A and Supplementary Fig. 3). Two doses of both vaccines elicited detectable IgG1 against WT SARS-CoV-2 Spike and Alpha and Beta VOC (Fig. 1B, left) that waned over time, resulting in more similar titers across the vaccine platforms at 4-5 months following immunization compared to peak immunogenicity. This dose-dependent increase and subsequent waning of receptor-binding domain (RBD) antibodies followed a similar trajectory (Fig. 1B, right). Other VOCs such as Gamma and Delta displayed similar patterns for IgG1 recognition against full-length Spike or RBD after vaccination with BNT162b2 or CoronaVac (Fig. 1C). This phenotype was specific to the SARS-CoV-2 VOCs, as demonstrated when we evaluated the antibody binding activity against the spike protein of seasonal human coronaviruses OC43 and HKU1, a seasonal Influenza HA and the Ebola glycoprotein (Supplementary Fig. 4). As expected, CoronaVac or BNT162b2 vaccination drove specific IgG1 responses against most VOCs, which also waned with time. Notably, Omicron Spike and RBD IgG1 responses were lower than responses observed for other VOCs in BNT162b2 and nearly undetectable in most CoronaVac recipients (Fig. 1C, bottom row).

### Distinct COVID-19 vaccines have unique Fc-receptor waning
To drive antibody-effector functions, antibodies must interact with Fc-receptors found on all innate immune cells[25,36,37]. Thus we next assessed whether vaccine-induced FcγR binding profiles, specifically the binding to the 4 human low-affinity Fcγ-receptors (FcγRIIA, FcγRIIB, FcγRIIIA, FcγRIIIB), waned with similar or disparate kinetics to antibody titers across the vaccine platforms. We observed distinct mRNA vaccine induced FcγR binding maturation and decay across the FcγRs. Specifically, FcγRIIA-binding Spike specific antibodies emerged rapidly after the first dose, peaking after the second dose, and remained detectable 4-5 months after vaccination, despite decaying to levels lower than observed after the first mRNA vaccine dose (Fig. 2A). Conversely, FcγRIIB, FcγRIIIA, and FcγRIIIB-binding Spike-specific levels exhibited peak responses after the second dose of the vaccine (Fig. 2B–D). These responses then declined rapidly over time, with nearly undetectable levels in most vaccinees 4-5 months following the completion of the primary mRNA vaccine series of immunization. These data point, for the first time, to differentiated waning across FcR-binding antibodies following mRNA vaccination, with the more robust persistence of FcγRIIA-binding Spike-specific antibodies over time.

Despite the clear induction of binding IgG antibodies by the CoronaVac vaccine, FcR binding profiles exhibited significant differences, both in peak titer and in waning kinetics. FcγRIIA- and FcγRIIIA-binding antibodies were induced most strongly by this vaccine in a dose-dependent manner. Strikingly, a single dose of the vaccine-induced low levels of these FcγR binding antibodies. FcγRIIA-binding Spike-specific antibodies accumulated after the second dose of the

vaccine but declined to near background levels 4–5 months following completion of the primary 2 dose series of immunizations. A subset of vaccinees further induced FcγRIIIA binding antibodies after the second dose of the vaccine, but all waned to undetectable levels. No FcγRIIB or FcγRIIIB-binding antibodies were induced by the first dose of the CoronaVac vaccine. Second-dose responses were present, but these quickly waned to background levels across all vaccinees (Fig. 2B, D) despite the clear induction of binding IgG antibodies (Fig. 1).

The same profiling was performed for RBD-specific antibodies across the 2 vaccine platforms. Again, BNT162b2 mRNA vaccination induced FcγRIIA-binding RBD-specific antibodies after a single dose, which matured exponentially after a second dose (Fig. 2E, left). However, these RBD-specific FcγRIIA-binding antibodies decayed rapidly over 4–5 months. Conversely, very low, although detectable, levels of RBD-specific FcγRIIA-binding antibodies were induced by the CoronaVac vaccine that fully decayed over 4–5 months (Fig. 2E, right).

Interestingly, 2 doses of BNT162b2 mRNA vaccination were able to drive RBD-specific antibodies capable to interact with FcγRIIB and FcγRIIIB binding antibodies, whereas a single dose produced low and heterogenous responses (Fig. 2F–H). Unlike FcγRIIA-binding antibodies, these waned to low-to-undetectable levels over 4-5 months. In contrast, the CoronaVac vaccination did not induce any RBD-binding antibodies able to interact with FcγRIIB or FcγRIIIB and only minimally produced FcγRIIIA binding antibodies, and these waned after 3 months. These data argue that both vaccines induced Spike and RBD-specific IgG1, despite showing distinct functional FcR binding properties. However, in general these functional properties waned more rapidly compared to the binding antibodies.

### mRNA vaccine boosts antibody and Fc effector functions
Given the comparatively low antibody levels induced by the CoronaVac vaccine against all VOCs, particularly Omicron, we next

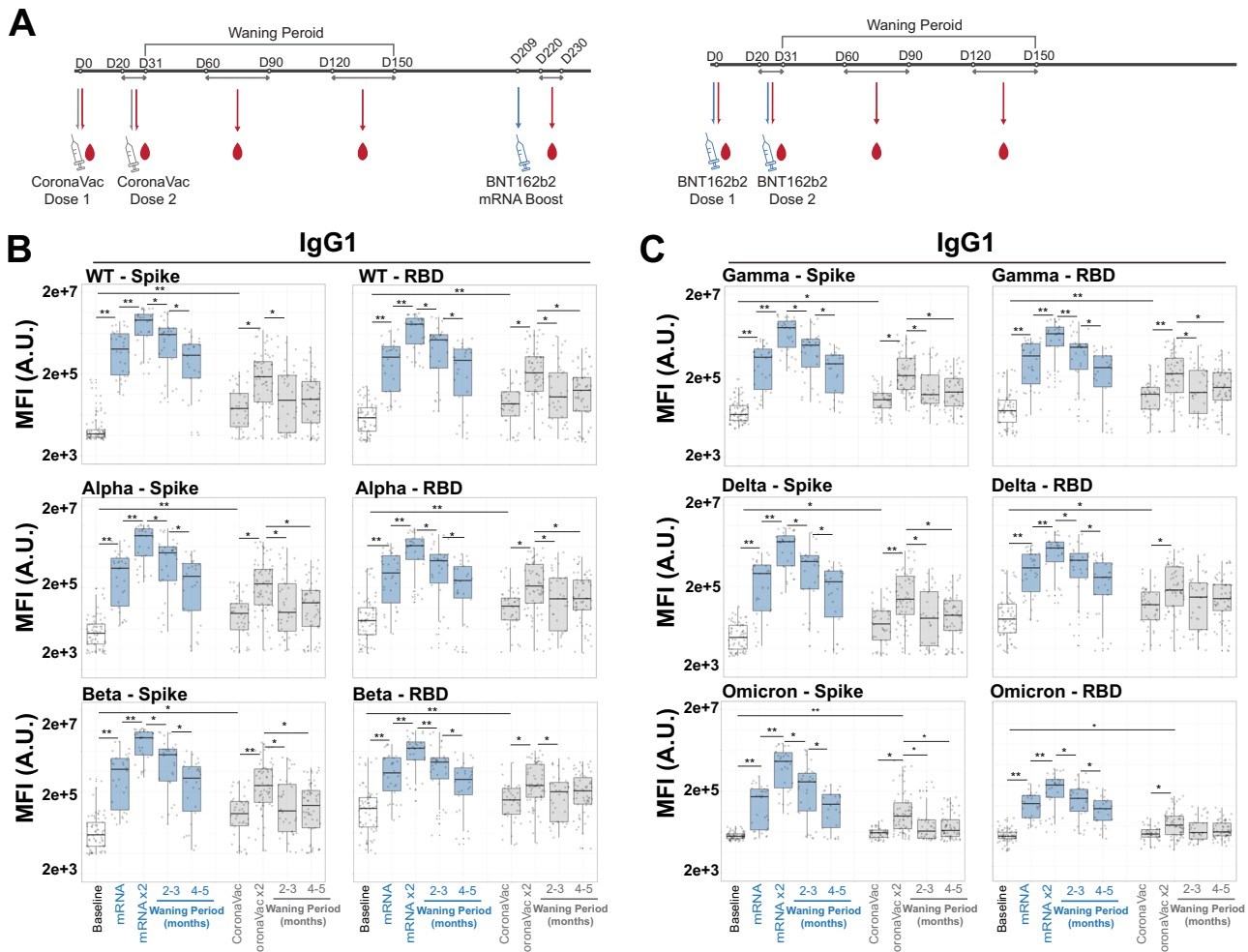

**Fig. 1 | COVID-19 vaccines exhibit a pan-variant waning of Spike-binding antibodies. A** Scheme for vaccination series and subsequent systems serology analysis. Participants who received two doses of the inactivated vaccine CoronaVac were boosted with the mRNA vaccine BNT162b2. As controls, the two-dose mRNA vaccine recipients were also analyzed. Sera were collected at various time points and systems serology assays were conducted (see Table 2 for specific days of sample collections). **B** Spike- (left) and receptor binding domain (RBD) (right) specific IgG1 levels for SARS-CoV-2 WT and early VOCs (Alpha and Beta) were quantified in the baseline (prior to immunization, white, lane 1), 1- and 2-dose BNT162b2 mRNA (blue, lanes 2 and 3, waning periods in lanes 4 and 5), 1- and 2-dose CoronaVac (gray, lanes 6 and 7, waning periods in lanes 8 and 9) via Luminex systems serology. Y-axis represents the MFI of binding in arbitrary units (A.U.) of a specific antigen. Shown

are box and whiskers, along with individual data points, which represent the mean of individual participants from each vaccine group (BNT162b2, n = 15 and CoronaVac, n = 34). The whisker above the box plot extends from the top quartile to the highest actual value that is within the 75th percentile + 1.5 * interquartile range. The whisker below the box plot extends from the lower quartile to the lowest actual value that is within 75th percentile + 1.5 * interquartile range. **C** Same as (**B**), but for latter VOC (Gamma, Delta, and Omicron) of SARS-CoV-2 VOC. All samples were assayed in technical duplicates. Kruskal–Wallis test was used for all panels (*p < 0.05 and **p < 0.01). All Kruskal–Wallis tests were two-sided, and no adjustments were made for multiple comparisons. See also Supplementary Fig. 3. Source data are provided as a Source Data file.

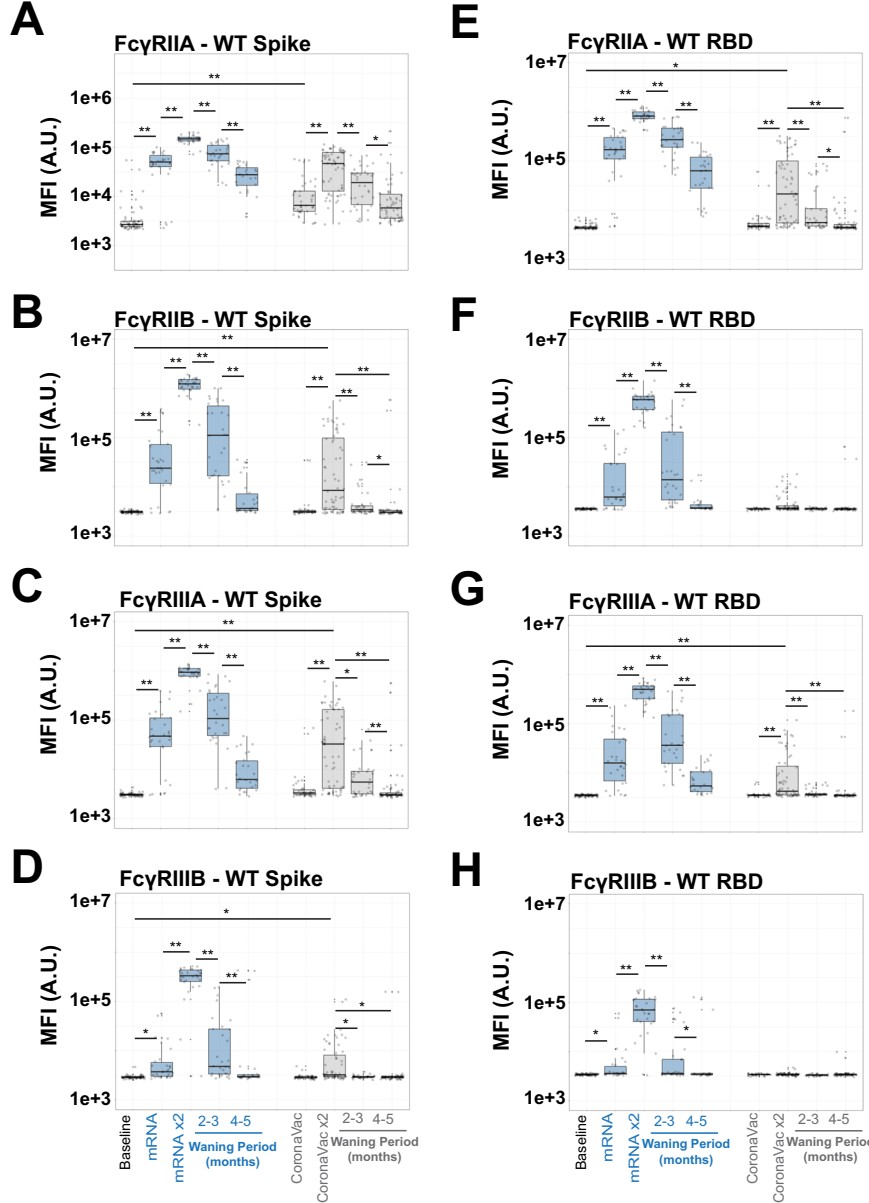

**Fig. 2 | FcR binding antibodies towards full-length Spike and RBD have rapid wane kinetics. A–D** FcR antibody binding towards WT SARS-CoV-2 Spike was quantified in the baseline (prior to immunization, white, lane 1), 1- and 2-dose BNT162b2 mRNA (blue, lanes 2 and 3, waning periods in lanes 4 and 5), 1- and 2-dose CoronaVac (gray, lanes 6 and 7, waning periods in lanes 8 and 9) via Luminex systems serology. Y-axis represents the MFI of binding full-length WT SARS-CoV-2 Spike. Shown are box and whiskers, along with individual data points, which represent the mean of individual participants from each vaccine group (BNT162b2, $n = 15$ and CoronaVac, $n = 34$). The whisker above the box plot extends from the top quartile to the highest actual value that is within the 75th percentile + 1.5 * interquartile range. The whisker below the box plot extends from the lower quartile to the lowest actual value that is within 25th percentile + 1.5 * interquartile range. (**A**) FcγRIIA, (**B**) FcγRIIB, (**C**) FcγRIIIA, and (**D**) FcγRIIIB. **E–H** Same as (**A–D**), but for SARS-CoV-2 RBD. Note that Y-axis scales are not the same. All samples were assayed in technical duplicates. Kruskal-Wallis test was used for all panels (*$p < 0.05$ and **$p < 0.01$). All Kruskal–Wallis tests were two-sided, and no adjustments were made for multiple comparisons. The samples were assayed in technical duplicates. See also Supplementary Fig. 4 for SARS-CoV-2 Nucleocapsid and control results. Source data are provided as a Source Data file.

investigated whether mRNA boosting of previous CoronaVac vaccinees could augment antibody breadth and Fc-effector function. Upon mRNA boosting, a sharp and pan-VOC increase in IgG1 levels was observed for full-length Spike (Fig. 3A) and for the RBD of all VOCs (Fig. 3B). This was particularly striking for Omicron-specific IgG responses, which exhibited the lowest cross-reactive IgG1 levels prior to boosting, yet Omicron-specific immunity was boosted to similar levels to other VOC Spike-specific responses (Fig. 3A), albeit Omicron RBD recognition was boosted but did not reach the levels of other RBD VOC responses.

To further examine the overall antibody isotype, subclass, and FcR binding breadth, we next examined the dynamics of the overall humoral profile of CoronaVac-induced SARS-CoV-2 specific antibodies over time and after boosting (Fig. 3C). As expected, WT Spike-specific responses had the highest and most consistent initial response but waned over the course of 5 months. This waning effect was not only rescued but also significantly boosted over peak titers following the Pfizer/BNT162b2 booster dose. This boost also raised highly functional IgG3- and IgM-responses to Spike, pointing to the induction of new responses. IgA responses evolved over time following CoronaVac

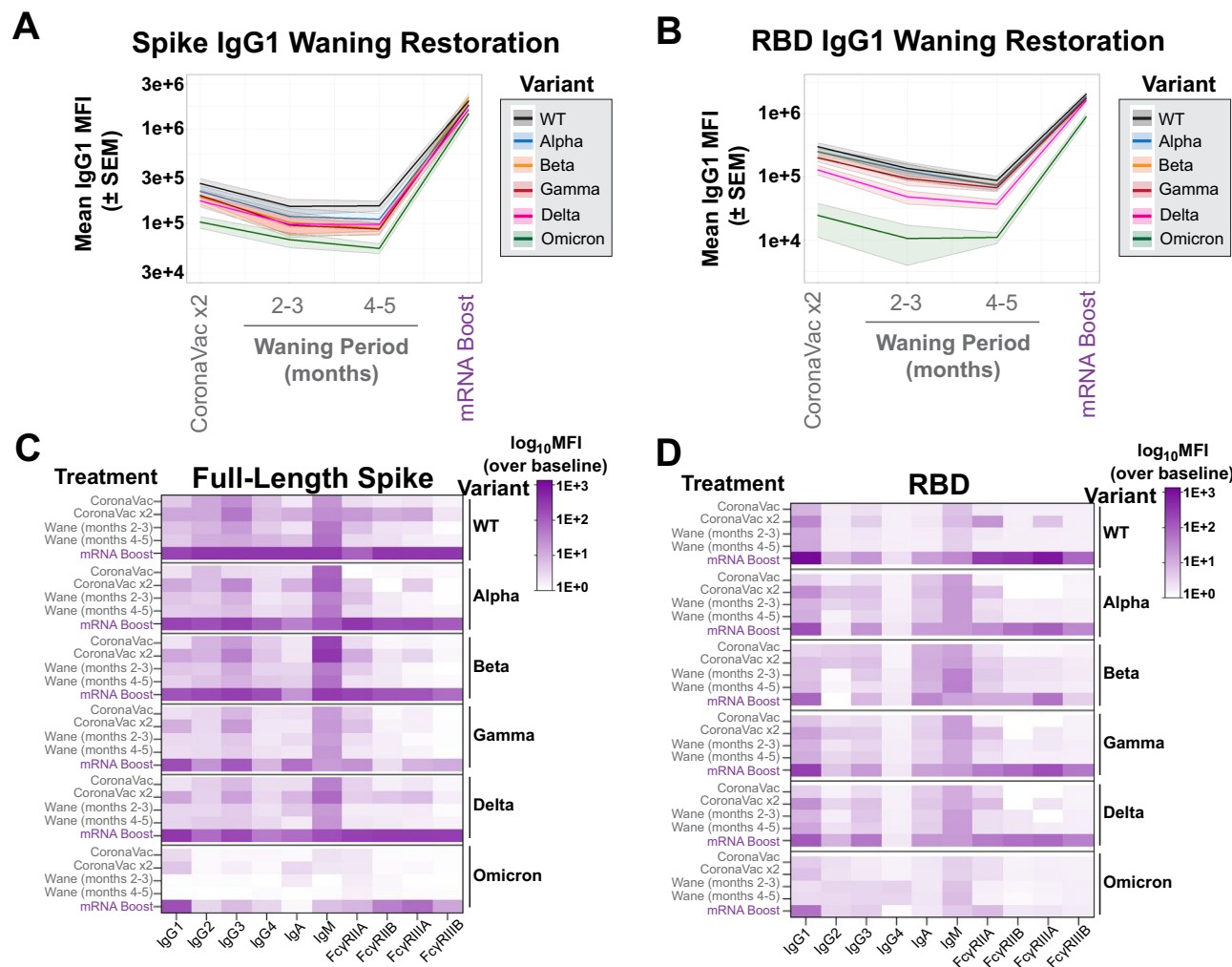

**Fig. 3 | COVID-19 mRNA boosters can enhance responses of multiple Spike-specific antibody-subclasses and -isotypes, and functional Fcγ-receptor complexes. A** Spike-specific IgG1 levels were measured at peak immunogenicity following the two-dose CoronaVac vaccination series (lane 1), during waning periods (lanes 2 and 3), and after mRNA boost (lane 4) for SARS-CoV-2 VOC (color legend shown on right). The Y-axis represents the MFI of the Spike from the VOC. Shown are means (solid line) with SEM for each VOC in the corresponding color in the shaded region. **B** Same as (**A**), but for the receptor-binding domains (RBD) of SARS-CoV-2 VOC. **C** Heatmap representation of binding for full-length Spike by antibody-subclasses and -isotypes and Fcγ-receptor complexes. Shown on the left are the description of CoronaVac doses, the subsequent waning period, and mRNA-vaccine booster, and on the right are the SARS-CoV-2 VOC or WT. The scale bar is shown to the right of the heatmap and represents MFI over baseline values. The values in each region represent the mean of values from individual participants from each vaccine group (BNT162b2, n = 15 and CoronaVac, n = 34). **D** Same as (**C**), but for the RBD of WT SARS-CoV-2 and VOC. The samples were assayed in technical duplicates. See also Supplementary Fig. 5. Source data are provided as a Source Data file.

immunization and were robustly boosted by the BNT162b2 response. Conversely, FcR-binding antibodies, key for eliciting antibody effector functions, were observed after the second CoronaVac dose and then quickly returned to near undetectable levels. However, following the BNT162b2 booster, all Spike-specific FcR binding responses were robustly enhanced against the WT Spike antigen.

Similar trends in subclass and isotype profiles were observed across VOCs. However, notably, Spike-specific IgA responses were less significantly induced to the Beta and Gamma-variants, potentially accounting for some potential mucosal transmission liabilities for this VOC. Additionally, IgG2 and IgG4 responses were induced weakly to the Gamma Spike. Omicron Spike-specific binding antibody FcR profiles differed most across the VOCs, with limited subclass and isotype responses to the Omicron Spike, despite robust IgG1 responses (Fig. 3C). Moreover, weaker FcγRIIA- and FcγRIIIB- Spike-specific binding antibodies were noted following the BNT162b2 boosting. In contrast, robust FcγRIIB and FcγRIIIA binding antibodies were induced, pointing to the selective induction of individual FcR binding responses to Omicron following boosting.

The cross-VOC RBD-specific response was more variable (Fig. 3D). While RBD-specific IgG1 responses were detected at nearly all time points following CoronaVac vaccination, BNT162b2 boosting resulted in a robust augmentation of RBD-specific IgG1 responses, superior to those observed at peak immunogenicity after the primary doses. Limited subclass and isotype responses were observed to the wildtype RBD with CoronaVac immunization alone. However, the addition of the mRNA booster significantly raised RBD-specific IgG3, IgA, IgM, and all FcR binding responses. A similar profile was observed to the Alpha, Gamma, and Delta VOC RBDs. However, the mRNA vaccine boosts only induced cross-reactive Beta RBD-specific antibodies with a more limited FcR binding profile, and a preferential significantly higher FcγRIIIA binding profile. Interestingly, while cross-reactive Omicron RBD-specific IgG1 were induced with the BTN162b2 boost, the boost failed to induce Omicron RBD-specific antibodies of distinct subclasses, isotypes or with broad FcR binding profiles. These data suggest that mRNA boosting can broaden the subclass, isotype, and FcR binding profile across VOCs (Supplementary Fig. 5), but may only partially rescue FcR binding specifically to more distant VOCs,

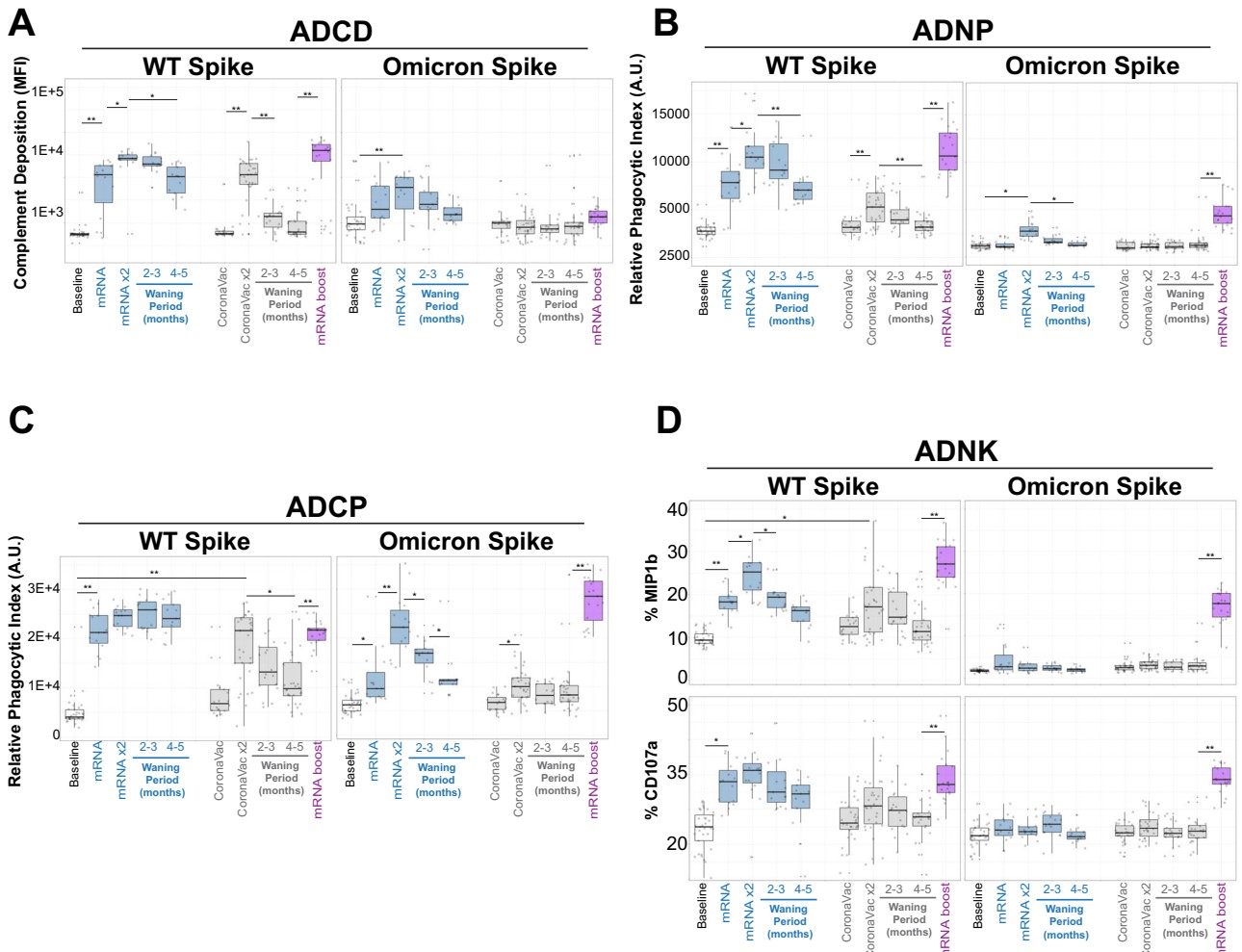

**Fig. 4 | mRNA-vaccine boosting of CoronaVac recipients broadens humoral defenses, including towards Omicron.** **A** (Left) Antibody-dependent complement deposition (ADCD) activities measured in fluorescent arbitrary units (A.U.) against the WT SARS-CoV-2 Spike (left) and Omicron Spike were quantified in the baseline (prior to immunization, white, lane 1), 1- and 2-dose BNT162b2 mRNA (blue, lanes 2 and 3, waning periods in lanes 4 and 5), 1- and 2-dose CoronaVac (gray, lanes 6 and 7, waning periods in lanes 8 and 9) and after mRNA-vaccine booster (lane 10). **B** Same as (**A**), but for antibody-dependent neutrophil phagocytosis (ADNP) measured in phagocytic A.U. **C** Same as (**A**) but for antibody-dependent monocytic cellular phagocytosis (ADCP) measured in phagocytic A.U. **D** Same as (**A**) but for the Primary natural killer (NK) cells activities measured for percent expression of MIP1b (top) and CD107a (bottom). Shown are box and whiskers, along with individual data points, which represent the mean of individual participants from each vaccine group (BNT162b2, $n = 15$ and CoronaVac, $n = 34$). The whisker above the box plot extends from the top quartile to the highest actual value that is within the 75th percentile + 1.5 * interquartile range. The whisker below the box plot extends from the lower quartile to the lowest actual value that is within 75th percentile + 1.5 * interquartile range. Note that Y-axis scales are not the same. All samples were assayed in technical quadruplicates with a minimum of three independent donors. Kruskal-Wallis test was used for all panels (*$p < 0.05$ and **$p < 0.01$). All Kruskal–Wallis tests were two-sided, and no adjustments were made for multiple comparisons. See also Supplementary Fig. 5. Source data are provided as a Source Data file.

particularly to those that have spread more efficiently and to which CoronaVac was shown to be slightly less efficient[38].

**mRNA-Vaccine boosting broadens functional humoral defenses**
While FcR binding is required to initiate Fc-effector function, we next sought to profile the longitudinal functional vaccine-induced humoral immune responses across the vaccine platforms, focused on the WT and Omicron SARS-CoV-2 Spike antigens. Although the CoronaVac vaccine elicited a moderate level of WT Spike-specific complement deposition (ADCD) (Fig. 4A, gray box plots) after the primary vaccine series, it quickly waned (64% and 72% reduction from peak activity at 2–3 months and 4–5 months from peak activity, respectively). The BNT162b2 vaccination elicited highly robust ADCD responses, that persisted for the first 2–3 months following the primary immunization series, but then the response waned at 4–5 months to a 58% overall reduction from peak activity (Fig. 4A, left). However, BNT162b2

boosting of CoronaVac vaccinees resulted in a highly significant expansion of ADCD responses, to levels higher than peak BNT162b2 mRNA-induced responses alone. Conversely, neither vaccine series elicited comparable immunity to the Omicron Spike (Fig. 4A, right), although mRNA-vaccine boosting was noted.

Similarly, WT Spike-specific antibody-dependent neutrophil phagocytosis (ADNP) was induced by both vaccines, albeit to lower levels with CoronaVac, although ADNP activities waned 30% and 36% after 4–5 months in CoronaVac and BNT162b2 recipients, respectively (Fig. 4B). However, after BNT162b2 boosting, ADNP levels in CoronaVac vaccinees rose to comparable levels to those observed after the primary series with the BNT162b2 vaccine. Importantly, the low-level Omicron-specific ADNP activity was noted after the second dose of the CoronaVac primary series, which declined rapidly. However, BNT162b2 boosting led to robust induction of Omicron-specific ADNP activity that was similar to WT Spike.

A nearly identical pattern was observed with antibody-dependent monocyte cellular phagocytosis (ADCP), albeit ADCP did not decline after the primary series of BNT162b2 vaccination for WT Spike (Fig. 4C). Again, while low Omicron ADCP was induced by the CoronaVac vaccine that waned, a cross-reactive response was induced with the BNT162b2 immunization boost, which was comparable to the WT-specific primary ADCP levels induced by the boost. Finally, WT-specific NK cell activation measured by NK cell degranulation through percent macrophage-inflammatory protein 1b (MIP-1b) expression (Fig. 4D, top) or percent surface expression of CD107a (Fig. 4D, bottom) positivity, was surprisingly elicited to comparable levels across CoronaVac and BNT162b2 vaccination after the primary series to the WT Spike antigens. Omicron-specific NK cell activating antibodies were only minimally observed among CoronaVac vaccinees after the primary series. However, after the BNT162b2 boost, both WT and Omicron-specific NK cell activating antibodies were significantly upregulated despite no activity being detected in the BNT162b2 recipients (Fig. 4D), pointing to a distinct functional maturation in the pre-existing setting of the CoronaVac recipients.

Given the emerging importance of antibody glycosylation as an important factor affecting the functional activities of vaccine-induced antibody responses against COVID-19[39,40], we next investigated the specific mechanism(s) by which the NK activity-enhancing antibodies was selectively induced in the boosted samples compared to the other groups. We conducted glycan analysis on samples from the naïve, at peak immunity, at the lowest decayed, and for the heterologous boosted timepoints (Supplementary Fig. 6). The relative abundances of N-glycan modifications were characterized, including the lack of fucose (afucosylation), bisecting N-acetyl glucosamine (GlcNAc), and galactose. Notably, anti-WT Spike IgG1 from individuals who received the first regimen of two doses of CoronaVac or BNT162b2 was reduced in core fucosylation when compared with the naïve group. Furthermore, those who received the third dose of heterologous boosting exhibited a further reduction in the level of fucosylation after the waning period (Supplementary Fig. 6A, B), in agreement with previous studies[41]. In addition, we also observed significant increases in bisecting glycans and galactosylation following vaccination and heterologous boosting (Supplementary Fig. 6C–F). Collectively these results indicate that primary vaccination and boosting have an impact in antibody glycosylation that modulates functional activities.

## Discussion

Emerging data suggest that distinct COVID-19 vaccination platforms can establish humoral responses with varying breath and magnitude towards Spike VOCs[4,24,38]. However, these responses wane rapidly across vaccine platforms[20,42]. Yet in addition to binding and neutralization, vaccine-induced antibodies are also capable of mediating an array of antibody effector functions[23,28], that have been linked to protection against severe disease and death[26] and monoclonal therapeutic activity[25,43]. However, whether these responses decay at the same rate as neutralizing antibody activity remains incompletely understood. A comparison of the 2 most widely distributed global COVID-19 vaccines, the CoronaVac and Pfizer/BioNTech mRNA vaccines revealed striking differences in peak antibody binding titers and Fc-effector functions across the 2 vaccine platforms, which both waned with time. Moreover, CoronaVac-induced Fc-effector functions demonstrated a steep and rapid decline to undetectable levels while binding to IgG was still present. These data suggest that the breadth of antibody binding does not directly translate to the level of circulating antibody effector function. Yet critically, a Pfizer/BioNTech boost of CoronaVac immunized recipients expanded the antibody effector function, in some cases above those observed with mRNA immunization alone. While the longitudinal duration of these boosted functional responses remains unclear, our results clearly show that effector

functions of antibodies are differentially induced and persist to different degrees across vaccine platforms during the primary vaccination series, and can be further tuned with boosting.

Previous work has shown that nearly all clinically approved COVID-19 vaccines elicit Spike-specific antibodies and can confer protection against wildtype virus induced disease[31,44–47]. The titers and functionalities of these antibodies vary across vaccine platforms[24,28] and wane with time[20,32,48,49]. We have previously shown that the two-dose vaccination scheme with CoronaVac induces neutralizing antibody titers to similar levels to those observed after a single dose of the BNT162b2 vaccine, but at significantly lower levels than those induced after two doses of this mRNA vaccine[50]. Here we show that this also extends to the Fc effector functions. We observed that following the primary series of CoronaVac vaccination, complement deposition, opsinophagocytic and NK cell activating antibodies were observed. However, many of these functions were induced to a slightly lower level than mRNA vaccination, with limited breadth to Omicron. mRNA vaccination, conversely, drove robust antibody effector functions. Nevertheless, many of these functions waned over time, including a loss of complement deposition, neutrophil phagocytosis, and NK cell degranulation. In contrast, monocyte phagocytosis remained stable over 4-5 months following mRNA vaccination, pointing to limited-to-no waning of this critical antibody effector function. Therefore, despite the general loss of neutralization and antibody effector functions that may result in a renewed susceptibility to infection, the persistence of some functional antibodies may continue to provide a first line of defense, potentially providing some level of long-term protection against the virus months after mRNA vaccination. Thus, similar to opsinophagocytic mechanisms of protection against other mucosal pathogens[51], it is plausible that persistent protection against severe disease and death afforded by mRNA vaccination may be linked to this opsinophagocytic activity.

In addition to the CoronaVac and Pfizer/BNT162b2 vaccines, previous studies have also suggested that heterologous boosting by other COVID-19 vaccines could potentially enhance the preexisting immune responses against the virus[52–54]. A vaccine boosting study demonstrated that seven different COVID-19 vaccines, including ChAdOx1 (Oxford-AstraZeneca), NVX-CoV2373 (Novavax), BNT162b2 (Pfizer–BioNTech), VLA2001 (Valneva), Ad26.COV2 (Janssen), mRNA1273 (Moderna), and CVnCoV (CureVac) are safe and induce strong immune responses when administered as boosters doses following two doses of either BNT162b2 or ChAdOx1 vaccines. Another study compared the safety and immunogenicity of a third heterologous booster dose of either the ChAdOx1, BNT162b2, or Ad26.COV2 vaccines in adults in Brazil who previously received two doses of CoronaVac. These results collectively indicated that a robust anamnestic immune response can be induced by each of these vaccines when used as a booster regardless of the primary COVID-19 vaccination regimen[55].

Previous meta-analysis have demonstrated that levels of IgG afucosylation, galactosylation, and bisecting GlcNAc differed in cohorts of COVID-19[39,40]. The level of bisecting GlcNAc was the most prominent feature distinguishing severe and mild COVID-19, where decreased bisection was found in severe patients. Higher levels of bisecting GlcNAc on IgG were reported to indirectly affect affinity for FcγRs and enhance ADCC by inhibiting fucosylation. Substantial changes in antibody galactosylation levels in severe COVID-19 infection result in a higher abundance of galactosylated IgG molecules compared to mild COVID-19, and such changes have been related to the pro-inflammatory effects of IgG through activation of the complement system. Previous studies reported decreased galactosylation both in anti-S IgG1 and total IgG1 in severe COVID-19 compared to mild, which is consistent with changes in the total IgG glycome[30]. Collectively, our results of glycan analysis are consistent with previous studies focusing on IgG glycome and COVID-19 infection.

**Table 1 | Demographic characteristics, and identification of SARS-CoV-2 infection cases**

|  | Overall (n = 49) | BNT162b2 (n = 15) | CoronaVac (n = 34) | Subgroup of CoronaVac (Booster dose of BNT162b2) (n = 23) |
|---|---|---|---|---|
| *Sex* | | | | |
| Male, *n* (%) | 13 (26.5%) | 3 (20%) | 10 (29.4%) | 8 (34.8%) |
| Female, *n* (%) | 36 (73.5%) | 12 (80%) | 24 (70.6%) | 15 (65.2%) |
| *Age* | | | | |
| 15–60 years, *n* (%) | 48 (98.0%) | 15 (100%) | 33 (97.1%) | 22 (95.7%) |
| Over 60 years, *n* (%) | 1 (2.0%) | – | 1 (2.9%) | 1 (4.3%) |
| Median (range) | 33 (15–80) | 36 (15–53) | 33 (21–80) | 35 (23–80) |
| *SARS-CoV-2 infection cases* | | | | |
| Participants, *n* (%) | 3 (6.1%) | – | 3 (8.8%) | 2 (8.7%) |

Despite the nearly complete loss of CoronaVac functional humoral immunity months after primary vaccination, we still observed a robust rise in CoronaVac primed immunity following an mRNA boost. Strikingly, this boosting was observed even for antibody functions that were not induced robustly following the primary vaccination series. Moreover, mRNA boosting also expanded the breadth of the response, arguing that the heterologous boosting strategy resulted in functional maturation for both the antigen-binding (Fab)- and constant (Fc)-domain-associated antibody responses. Whether similar patterns of functional maturation of the humoral immune responses would be observed in the presence of alternate mix-and-match strategies remains unclear, however; the data presented here strongly argue that combinations of these highly distributed vaccines could potentially be used in optimal combinations to drive broad pan-VOC functional immunity.

Analysis across VOCs highlighted significant differences in variant recognition by the FcR binding antibodies. Specifically, following the primary CoronaVac series, IgG1 antibodies exhibited some cross-VOC recognition; however, CoronaVac-induced subclasses/isotypes and FcR binding antibodies exhibited little-to-no RBD and Spike recognition across Beta, Gamma, and Omicron. Only the CoronaVac-induced IgM responses exhibited cross VOC immunity. However, upon mRNA boosting, all antibody sucblass/isotypes/FcR binding increased to levels comparable to those against the WT spike, but subclass and FcR binding elevated to a lesser extent against the Gamma and Omicron Spike variants. Similarly, boosting drove robust FcR binding to the WT RBD, but more limited FcR binding to the Omicron RBD, despite strong binding antibodies to this variant antigen. Thus, these data suggest that binding to the RBD may not always be proportional to FcR binding and that other characteristics, including geometry and stoichemiotry, may also play an essential role in dictating antibody effector functions. Hence, further improvements in the quality of immunity may be achievable with future mix-and-match strategies, aimed at eliciting the strongest level of protective immunity against future VOCs. It is noteworthy that the measurements shown here for the IgG subclassing and receptor binding were independently performed. Therefore, the antibody Fc-mediated binding activities are not assigned to an IgG subclass. Future studies on the relationship between the high-affinity FcγRI and antibody functional responses could potentially further clarify the overall humoral responses towards COVID-19.

Whether the heterologous nature of the mRNA boost or the timing of the delayed boosting drove improved CoronaVac Fab and Fc activity remains unclear. Yet, emerging data suggest that both the nature of the platform and the timing of immunization play a key role in the quality of the germinal center response, and thus humoral immune induction[52,55–57]. Consequently, future studies are warranted to begin to probe the mix-and-match and interval based effects of vaccine induced immune programming to help guide future rational vaccine campaigns aimed at driving the most protective vaccine induced immunity against SARS-CoV-2 and future variants.

Studies have shown that the neutralizing activity of vaccine-elicited antibodies could be more focused on the receptor-binding domain (RBD) as compared to infection-elicited antibodies. However, within the RBD, the binding of vaccine-elicited antibodies is more broadly distributed across epitopes than those seen for infection-elicited antibodies[58]. The rapid decay of RBD-specific antibodies could be due to the fact that RBD is a smaller sub-domain of the Spike protein. Therefore, targeted waning could be more rapid for RBD than for the full-length Spike, independent of the vaccine platform. This greater binding breadth suggests that single RBD mutations might have less impact on neutralization by vaccine sera than convalescent sera. Hence, antibody immunity acquired by different means may have differing susceptibility to erosion by viral evolution[59–61]. In addition, the significant variability of individual responses is considered a difficult confounder in these types of studies. We thus, deliberately grouped samples together to account for heterogeneity between individual responses. Therefore, statistical grouping based on this method should be considered as conservative approach given the known discrepancy in these responses.

The CoronaVac vaccine is one of the most widely distributed vaccine globally, with billions of doses administered worldwide[33,55,56]. While the vaccine may not induce the most robust neutralizing antibody titers, protection against severe disease and death persisted across several VOCs[33]. Moreover, similar to other vaccines based on whole viral particles, this vaccine gives exposure to all viral components, which may provide additional protection in the setting of future viral variation.

## Methods
### Resource availability
**Lead contact.** Further information and requests for resources and reagents should be directed to and will be fulfilled by the lead data point of contact, Ryan P. McNamara (rpmcnamara@mgh.harvard.edu), or the corresponding authors, Galit Alter (galter@mgh.harvard.edu), or Rafael A. Medina (rafael.medina@emory.edu).

### Experimental model and participants details
Serum samples were obtained from participants who received the complete-dosage regimen of the respective vaccines as recommended by the manufacturers. The cohort contains samples from individuals who received either BNT162b2 (*n* = 15) or CoronaVac vaccines (*n* = 34) (Tables 1, 2). The BNT162b2 mRNA vaccine group was given 30 μg BNT162b2 (15–53 years old, median: 36 years, 80% female) on days 0 and 21, and serum samples were taken up to 168 days after the second dose. The CoronaVac group (21–80 years old, median: 33 years old, 70.6% female) received two doses of 600 U CoronaVac four weeks apart, and individuals were sampled up to 209 days after the second dose. A subgroup of CoronaVac participants (*n* = 23, 23–80 years old, median: 35 years old, 65.2% female) received a booster dose of the BNT162b2 mRNA vaccine and were sampled 14–31 days after the mRNA

**Table 2 | Specific time intervals (days) of samples obtained from individuals vaccinated with either CoronaVac or BNT162b2**

| Participants number[a] | Vaccine | Pre-Vaccine sample dose 0 (n = 33) | Vaccine sample dose 1 (n = X) | Vaccine sample dose 2 (n = X) | Waning period sample 2–3 months (n = X) | Waning period sample 4–5 months (n = X) | Vaccine sample BNT162b2 booster (n = 23) |
|---|---|---|---|---|---|---|---|
| 1 | CoronaVac | 0 | 29 | 41 | 104 | 153 | 231 |
|  |  |  | 1 | 13 | 76 | 125 | 203 |
|  |  |  |  | 14 |  |  |  |
| 2 | BNT162b2 | 0 | 20 | 34 | 105 | 181 |  |
|  |  |  |  | 13 | 84 | 160 |  |
| 3 | BNT162b2 | 0 | 20 | 35 | 104 | 181 |  |
|  |  |  |  | 13 | 82 | 159 |  |
| 4 | BNT162b2 | 0 | 20 | 35 | 104 | 188 |  |
|  |  |  |  | 14 | 83 | 167 |  |
| 5 | BNT162b2 | 0 | 21 | 34 | 113 | 181 |  |
|  |  |  |  | 13 | 92 | 160 |  |
| 6 | BNT162b2 | 0 | 20 | 36 | 104 | 188 |  |
|  |  |  |  | 14 | 82 | 166 |  |
| 7 | BNT162b2 | 0 | 21 | 34 | 104 | 189 |  |
|  |  |  |  | 13 | 83 | 168 |  |
| 8 | BNT162b2 | 0 | 20 | 40 | 104 | 188 |  |
|  |  |  |  | 19 | 83 | 167 |  |
| 9 | BNT162b2 | 0 | 20 | 36 | 104 | 179 |  |
|  |  |  |  | 15 | 83 | 158 |  |
| 10 | BNT162b2 | 0 | 20 | 36 | 104 | 188 |  |
|  |  |  |  | 15 | 83 | 167 |  |
| 11 | BNT162b2 | 0 | 20 | 40 | 104 | 188 |  |
|  |  |  |  | 19 | 83 | 167 |  |
| 12 | BNT162b2 | 0 | 20 | 34 | 124 | 180 |  |
|  |  |  |  | 14 | 104 | 160 |  |
| 13 | CoronaVac | −9 | 28 | 47 | 102 | 174 |  |
|  |  |  | 0 | 19 | 74 | 146 |  |
| 14 | CoronaVac | −8 | 27 | 42 | 98 | 175 |  |
|  |  |  |  | 13 | 69 | 146 |  |
| 15 | CoronaVac | −7 | 30 | 43 |  |  |  |
|  |  |  | 2 | 15 |  |  |  |
| 16 | CoronaVac | −7 | 28 | 42 | 105 |  |  |
|  |  |  | 0 | 14 | 77 |  |  |
| 17 | CoronaVac | −8 | 28 | 41 | 105 | 180 | 232 |
|  |  |  | 0 | 13 | 77 | 152 | 204 |
|  |  |  |  |  |  |  | 16 |
| 18 | CoronaVac | −6 | 30 | 42 | 99 | 175 | 232 |
|  |  |  | 2 | 14 | 71 | 147 | 204 |
|  |  |  |  |  |  |  | 14 |
| 19 | CoronaVac | 0 | 28 | 50 |  |  |  |
|  |  |  | 0 | 22 |  |  |  |
| 20 | CoronaVac | 0 | 29 | 41 | 104 | 174 | 232 |
|  |  |  | 1 | 13 | 76 | 146 | 204 |
|  |  |  |  |  |  |  | 14 |
| 21 | CoronaVac | −6 | 28 | 42 | 91 | 181 | 233 |
|  |  |  | 0 | 14 | 63 | 153 | 205 |
|  |  |  |  |  |  |  | 14 |
| 22 | CoronaVac | −2 | 27 | 42 | 96 | 180 | 237 |
|  |  |  |  | 14 | 68 | 152 | 209 |
|  |  |  |  |  |  |  | 14 |
| 23 | CoronaVac | −7 | 28 | 43 | 91 | 175 | 231 |
|  |  |  | 0 | 15 | 63 | 147 | 203 |
|  |  |  |  |  |  |  | 15 |

**Table 2 (continued) | Specific time intervals (days) of samples obtained from individuals vaccinated with either CoronaVac or BNT162b2**

| Participants number[a] | Vaccine | Pre-Vaccine sample dose 0 ($n=33$) | Vaccine sample dose 1 ($n=X$) | Vaccine sample dose 2 ($n=X$) | Waning period sample 2–3 months ($n=X$) | Waning period sample 4–5 months ($n=X$) | Vaccine sample BNT162b2 booster ($n=23$) |
|---|---|---|---|---|---|---|---|
| 24 | CoronaVac | 0 | 28 | 42 | 96 | 173 | |
| | | | −1 | 13 | 67 | 144 | |
| 25 | CoronaVac | −4 | 28 | 43 | 91 | 157 | |
| | | | 0 | 15 | 63 | 129 | |
| 26 | CoronaVac | −1 | 27 | 42 | 95 | 173 | 229 |
| | | | | 13 | 66 | 144 | 200 |
| | | | | | | | 14 |
| 27 | CoronaVac | −4 | 28 | 42 | 87 | 167 | 233 |
| | | | 0 | 14 | 59 | 139 | 205 |
| | | | | | | | 15 |
| 28 | CoronaVac | 0 | 28 | 42 | 90 | 167 | |
| | | | 0 | 14 | 62 | 139 | |
| 29 | CoronaVac | 0 | 28 | 42 | 89 | 167 | 232 |
| | | | 0 | 14 | 61 | 139 | 204 |
| | | | | | | | 14 |
| 30 | CoronaVac | 0 | 29 | 42 | 91 | 188 | |
| | | | 1 | 14 | 63 | 160 | |
| 31 | BNT162b2 | 0 | 26 | 39 | 112 | 182 | |
| | | | 1 | 14 | 87 | 157 | |
| 32 | CoronaVac | | | 54 | | 148 | 229 |
| | | | | 26 | | 120 | 201 |
| | | | | | | | 14 |
| 33 | CoronaVac | | | 54 | | 166 | 231 |
| | | | | 26 | | 138 | 203 |
| | | | | | | | 16 |
| 34 | CoronaVac | | | 55 | | 149 | 231 |
| | | | | 27 | | 121 | 203 |
| | | | | | | | 14 |
| 35[b] | CoronaVac | | | 52 | | 146 | |
| | | | | 24 | | 118 | |
| 36 | CoronaVac | | | 53 | | 151 | 230 |
| | | | | 25 | | 123 | 202 |
| | | | | | | | 14 |
| 37[b] | CoronaVac | | | 59 | | 153 | 231 |
| | | | | 31 | | 125 | 203 |
| | | | | | | | 18 |
| 38 | CoronaVac | | | 59 | | 153 | 230 |
| | | | | 31 | | 125 | 202 |
| | | | | | | | 15 |
| 39 | CoronaVac | | | 49 | | 143 | 224 |
| | | | | 21 | | 115 | 196 |
| | | | | | | | 14 |
| 40 | CoronaVac | | | 55 | | 149 | 230 |
| | | | | 27 | | 121 | 202 |
| | | | | | | | 31 |
| 41 | CoronaVac | | | 56 | | 150 | 231 |
| | | | | 28 | | 122 | 203 |
| | | | | | | | 14 |
| 42 | CoronaVac | | | 59 | | 158 | 235 |
| | | | | 31 | | 130 | 207 |
| | | | | | | | 19 |

**Table 2 (continued) | Specific time intervals (days) of samples obtained from individuals vaccinated with either CoronaVac or BNT162b2**

| Participants number[a] | Vaccine | Pre-Vaccine sample dose 0 (n = 33) | Vaccine sample dose 1 (n = X) | Vaccine sample dose 2 (n = X) | Waning period sample 2–3 months (n = X) | Waning period sample 4–5 months (n = X) | Vaccine sample BNT162b2 booster (n = 23) |
|---|---|---|---|---|---|---|---|
| **43** | CoronaVac | | | 59 | | 153 | 230 |
| | | | | 31 | | 125 | 202 |
| | | | | | | | 15 |
| **44** | CoronaVac | | | 56 | | 154 | 233 |
| | | | | 28 | | 126 | 205 |
| | | | | | | | 14 |
| **45** | BNT162b2 | 10 | | 42 | 91 | 181 | |
| | | | | 14 | 63 | 153 | |
| **46[c]** | CoronaVac | | | 33 | 127 | | |
| | | | | 6 | 100 | | |
| **47** | CoronaVac | | | 58 | | 149 | 232 |
| | | | | 30 | | 121 | 204 |
| | | | | | | | 14 |
| **48** | BNT162b2 | 1 | 27 | 42 | 90 | | |
| | | | | 14 | 62 | | |
| **49** | BNT162b2 | −1 | 26 | 40 | 96 | | |
| | | | | 13 | 69 | | |

[a]All individuals in this cohort were individuals without prior COVID-19 vaccination.
[b]SARS-CoV-2 infection after the second dose of vaccine.
[c]SARS-CoV-2 infection after the first dose of vaccine.

booster. We did not observe any immunocompromising comorbidities associated with the cohort. All the individuals included in the study were naïve at the time of vaccination. Previous exposure to SARS-CoV-2 was ruled out by a qRT-PCR test at the time of recruitment. Only 3 individuals reported to be exposed to COVID-19 between vaccine dose 1 and 2 (1 individual) or after the second dose (2 individuals). In addition, we monitored for infection-acquired anti-nucleocapsid antibodies throughout the study. Since CoronaVac contains nucleocapsid, a noticeable and expected response was observed exclusively within this group. Moreover, nucleocapsid antibodies waned progressively over five months within this group, arguing against any exposure to SARS-CoV2 during this period. No nucleocapsid responses were observed in the BNT162b2 arm as this mRNA vaccine only encodes for Spike protein. We thus collectively conclude that, SARS-CoV-2 infections prior, before and during this study were minimal-to-absent (Supplementary Fig. 4A).

**Method details**

**Antigens.** All antigens used in this study are listed in Table 3. Most antigens were lyophilized powder and were resuspended in water to afford a final concentration of 0.5 mg/mL. Antigens that required biotinylation were treated with the NHS-Sulfo-LC-LC kit per the manufacturer's instruction. Removal of excess biotin and buffer exchange from Tris-containing antigens was done using the Zebra-Spin desalting and size exclusion chromatography columns.

**Immunoglobulin isotype and Fc receptor binding.** Antigen-specific antibody levels of isotypes and subclasses and levels of Fcγ-receptor binding were evaluated using a custom multiplexing Luminex-based assay platform in technical replicates, as previously described[36]. The general sensitivity and robustness of the assays used in this study to quantify the antibody titers and relative FcγR binding capacity and specifically in the context of SARS-CoV-2 infections and vaccines have been previously validated[24,62–64]. The sensitivity of the assays used specifically to evaluate the antibody isotypes and the receptor binding was also validated by the dilution curves generated prior to the

analysis of the samples, as shown in Supplementary Fig. 2. The antigens were directly coupled to magnetic Luminex beads (Luminex Corp, TX, USA) by carbodiimide-NHS ester-coupling chemistry, which designates each region to each antigen. Individual dilution curves for each antigen were performed to identify an appropriate dilution factor for each secondary feature that was within the linear range of detection. The antigen-coupled beads were incubated with different serum dilutions (1:100 for IgG2, IgG3, IgG4, IgM, and IgA1, 1:500 for IgG1, and 1:750 for Fcγ-receptor binding) overnight at 4 °C in 384 well plates (Greiner Bio-One, Germany). Unbound antibodies were removed by washing and subclasses, isotypes were detected using the respective PE-conjugated antibody listed in Table 3. All detection antibodies were used at a 1:100 dilution. All samples were assayed in technical duplicates.

For the analysis of Fcγ-receptor binding PE-Streptavidin (Agilent Technologies, CA, USA) was coupled to recombinant and biotinylated human FcγRIIA, FcγRIIB, FcγRIIIA, or FcγRIIIB proteins. Coupled Fcγ-receptors were used as a secondary probe at a 1:1000 dilution. After 1 h of incubation, the excessive secondary reagent was removed by washing, and the relative antibody concentration per antigen was determined on an IQue Screener PLUS cytometer (IntelliCyt).

**Evaluation of antibody-mediated functions.** For all the antibody-mediated functional assays performed in this study, only two antigens, SARS-CoV-2 WT Spike (Sino Biological, 40589-V08H4) and SARS-CoV-2 Omicron Variant S (Sino Biological, 40589-V08H26) were used as discussed in detail below. All samples were assayed in technical quadruplicates with a minimum of three independent donors.

Antibody-dependent cellular phagocytosis (ADCP) and neutrophil phagocytosis (ADNP) were evaluated using a flow cytometry-based phagocytic assay that requires the usage of fluorescently labeled microspheres, as described previously[36]. The WT and Omicron Spike antigens were biotinylated and conjugated to yellow-green fluorescent neutravidin microspheres. Then diluted serum samples with the pre-determined concentrations (1:100) were incubated with the coupled antigens. The pre-formed immune

## Table 3 | List of reagents and resources used in this study

| REAGENT or RESOURCE | SOURCE | IDENTIFIER |
|---|---|---|
| Anti-Human IgG1-PE | Southern Biotech | HP6001 |
| Anti-human IgG2-PE | Southern Biotech | 31-7-4 |
| Anti-human IgG3-PE | Southern Biotech | HP6050 |
| Anti-human IgG4-PE | Southern Biotech | HP6025 |
| Anti-human IgM-PE | Southern Biotech | SA-DA4 |
| Anti-human IgA1-PE | Southern Biotech | HP6025 |
| Anti-CD66b Pac Blue | BioLegend | 305112 |
| Anti-CD107a | BD Biosciences | 555802 |
| Anti-CD3 | BD Biosciences | 558117 |
| Anti-CD16 | BD Biosciences | 557758 |
| Anti-CD56 | BD Biosciences | 557747 |
| Anti-IFNγ | BD Biosciences | 340449 |
| Anti-CCL4 | BD Biosciences | 550078 |
| Anti-C3b | MP Biomed | 855385 |
| SARS-CoV-2 WT Spike | Sino Biological | 40589-V08H4 |
| SARS-CoV-2 WT Receptor Binding Domain (RBD) | Sino Biological | 40592-V08H |
| SARS-CoV-2 Alpha Variant S | Sino Biological | 40589-V08B6 |
| SARS-CoV-2 Alpha Variant RBD | Sino Biological | 40592-V08H82 |
| SARS-CoV-2 Beta Variant S | Sino Biological | 40589-V08B7 |
| SARS-CoV-2 Beta Variant RBD | Sino Biological | 40592-V08H59 |
| SARS-CoV-2 Gamma Variant S | Sino Biological | 40589-V08B10 |
| SARS-CoV-2 Gamma Variant RBD | Sino Biological | 40592-V08H86 |
| SARS-CoV-2 Delta Variant S | Sino Biological | 40589-V08B16 |
| SARS-CoV-2 Delta Variant RBD | Sino Biological | 40592-V08H115 |
| SARS-CoV-2 Omicron Variant S | Sino Biological | 40589-V08H26 |
| SARS-CoV-2 Nucleocapsid | Sino Biological | 40588-V08B |
| SARS-CoV-2 Omicron Variant RBD | Sino Biological | 40592-V08H121 |
| Human Coronavirus OC43 S | Sino Biological | 40607-V08B |
| Human CoV HKU1 S (isolate N5) | Sino Biological | 40606-V08B |
| Human Cytomegalovirus (HCMV) Glycoprotein B (gB) | Sino Biological | 10202-V08H1 |
| Ebola Virus Glyoprotein | IBT Bioservices | 0501-015 |
| PE-Streptavidin | Agilent Technologies | PB32-10 |
| Guinea Pig Complement | Cedarlane | CL4051 |
| Protein Transport Inhibitor | BD Biosciences | 554724 |
| Brefeldin-A | Sigma | B7651 |
| NHS-Sulfo-LC-LC Kit | ThermoFisher | 21435 |
| Zebra-Spin Desalting and Chromatography Columns | ThermoFisher | 89882 |
| RosetteSep NK Enrichment Kit | Stem Cell Technologies | 15065 |
| Fix & Perm Cell Permeabilization Kit | ThermoFisher | GAS002S-100 |
| R Studio V 1.4.1103 | RStudio, PBC | Open Source |
| GraphPad Prism v 9.3.1 | GraphPad Software, LLC | Ragon Site License |
| FlowJo V. 10.8 | FlowJo, LLC | www.flowjo.com/ solutions/flowjo/ downloads |
| iQue Forecyt 9.1 | Sartorius | 60028 |
| iQue Screener Plus | Intellicyt/Sartorius | 11811 |
| 384-well HydroSpeed Plate Washer | Tecan | 30190112 |
| MagPlex Microspheres | Luminex MFG | MC12001-01 (Cataloged by region) |
| Green Fluorescent Neutravidin Microspheres | ThermoFisher | F8776 |
| Red Fluorescent Neutravidin Microspheres | ThermoFisher | F8775 |

complexes bound with microspheres were washed and incubated with a human monocyte cell line (THP-1) for ADCP function or with neutrophils collected from healthy donors' blood samples to assess ADNP activity. Cells studied under ADNP assays were then stained with anti-CD66b Pac blue antibody to calculate the percentage of CD66b+ neutrophils. In both ADCP and ADNP assays cells were fixed with 4% paraformaldehyde (PFA) and identified by gating on single cells and microsphere-positive cells. Microsphere uptake was quantified as a phagocytosis score, calculated as the (percentage of microsphere-positive cells) x (MFI of microsphere-positive cells) divided by 100,000.

Antibody-dependent complement deposition (ADCD) assessed the ability of antigen-specific 20 antibodies to bind complement component C3b, as described. Briefly, SARS-CoV-2 WT and Omicron Spike proteins were biotinylated, coupled to red fluorescent neutravidin microspheres, and then incubated with serum samples (dilution 1:40). Immune complexes were then washed, incubated with guinea pig complement, then washed with 15 mM EDTA. The level of complement deposition was measured by fluorescein-conjugated goat IgG that targets the guinea pig complement C3b and further analyzed by gating on the single microspheres and C3b+ events.

Antibody-dependent NK Cell activation (ADNKA) assessed the ability of antigen-specific antibodies to activate human NK cells to up-regulate the production of CD107a, IFN-γ, and CCL4 (MIP-1β), as described. Briefly, ELISA plates were coated with SARS-CoV-2 antigens. NK cells were isolated from buffy coats, obtained from healthy blood donors, using RosetteSep NK enrichment kit, and rested overnight with IL-15. Antigen-coated ELISA plates were then incubated with serum samples (dilution 1:10). NK cells were stained with anti-CD107a and treated with a protein-transport inhibitor and with brefeldin A to block degranulation. NK cells were then added to the immune complexes, labeled with surface staining antibodies including anti-CD3, anti-CD16, and anti-CD56, washed, fixed with 4% PFA, and permeabilized with PERM A/B to allow intracellular staining with anti-IFN-γ and anti-CCL4. NK cells were identified as CD3- CD56+ cells, and the level of NK cell activation was evaluated as CD107a+ IIFN-γ + CCL4 + .

**Glycosylation analysis.** Antigen-specific glycosylation analysis was performed as described previously[65] Briefly, the WT Spike antigen was biotinylated and coupled to NeutrAvidin magnetic beads at a ratio of 2.5 μg of protein to 25 μL of beads. Combined serum (200 μL) was first incubated with non-coated NeutrAvidin beads to remove the non-specific binding and then added to the antigen-coated beads and incubated for 1 h at 37 °C. Antibodies were eluted by incubation in 50 μL of pH 2.9 citrate buffer for 30 min at 37 C. Samples were then spun down and the eluted antibodies, contained in the supernatant, were neutralized with 30 μL pH 8.9 potassium phosphate buffer. Antibodies were then coupled to protein G beads. After the beads were washed, IDEZ protease was used to cleave the Fab (in the supernatant) from the Fc (remained on the magnetic beads) for 1 h at 37 °C and collected. The two fragments were deglycosylated and fluorescently labeled using a GlycanAssure APTS kit. For the Fc fragment, which remained bound to the protein G beads, an additional magnetic separation after PNGase glycan cleavage separated the glycans from the remaining magnetic beads and the protocol then proceeded. Glycans were analyzed on a 3500xL genetic analyzer. Glycan fucosyl and afucosyl libraries were used to assign 24 discrete glycan peaks using GlycanAssure software. Data are reported as percentages of total glycans for each of the glycan peaks.

### Statistics and reproducibility
All data analysis was done using R Studio V 1.4.1103 or FlowJo. Statistical analysis was done using R studio or GraphPad Prism v 9.3.1. Box and whisker plots were generated using ggplot, calculating the mean and standard deviation for each factor. Technical replicates for each sample were performed and the mean between replicates was plotted. No statistical method was applied to predetermine the size of the cohort that were evaluated in this study. No data was omitted from the analysis. The investigators who performed the experiments were

blinded before and while conducting the reported experiments. The samples were randomized for all experimental procedures and the investigators were unblinded only after the data were collected.

### Inclusion and ethics statement
Healthy individuals that received their COVID-19 vaccine and booster immunizations at the UC-Christus Health network in Santiago, Chile, were invited to participate in the study. There was no specific selection of participants and they represent the overall population vaccinated in the UC-Christus Health Network between January and September, 2021. An informed written consent form was obtained under protocol 200829003, which was reviewed and approved by the Health Sciences Scientific Ethics Committee at Pontificia Universidad Católica de Chile (PUC). This work was supervised and approved by the Mass General Institutional Review Board (IRB #2020P00955 and #2021P002628).

### Reporting summary
Further information on research design is available in the Nature Portfolio Reporting Summary linked to this article.

## Data availability
All the data generated in this study are provided within the article and the Supplementary data is provided as a Source Data file. The Systems Serology data generated in this study are availbale in the ImmPort database under accession code SDY2216. Source data are provided with this paper.

## Code availability
All coding was done using R Studio V 1.4.1103 using ggplot. Individual groups were analyzed as factors. All codes and scripts are available upon request to the lead data point of contact. This article does not report original code. Any additional information required to reanalyze the data reported in this article is available from the lead contact upon request.

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

## Acknowledgements

The authors would like to thank the laboratory of Prof. Douglas Lauffenburger (Massachusetts Institute of Technology) for the critical evaluation of statistics in this manuscript. We also thank Mark and Lisa Schwartz, Terry and Susan Ragon, and the SAMANA Kay MGH Research Scholars award for their support. GA receives funding from the Massachusetts Consortium on Pathogen Readiness (MassCPR), the Gates Global Health Vaccine Accelerator Platform, and the NIH (3R37AI080289-11S1, R01AI146785, U19AI42790-01, U19AI135995-02, U19AI42790-01, P01AI1650721, U01CA260476-01). We also would like to thank Estefany Poblete, Erick Salinas and Andres Muñoz for their excellent technical and professional expertise during clinical recruitment and sample processing. Work in the Medina laboratory was partially funded by the FONDECYT 1212023 grant from ANID of Chile, the FLUOMICS and SYBIL Consortium (NIH-NIAID grant U19AI135972) and the Center for Research on Influenza Pathogenesis (CRIP), an NIAID Center of Excellence for Influenza Research and Surveillance (CEIRS, contract # HHSN272201400008C) awarded to RAM. CPR and JL conducted this work as part of their Postdoctoral grant FONDECYT 3190706 and 3190648, respectively. MJA and ES conducted this work as part of their Ph.D. Thesis, under Programa de Doctorado en Ciencias Biológicas mención Genética Molecular y Microbiología, Facultad de Ciencias Biológicas, Pontificia Universidad Cátolica de Chile. MJA was funded by the ANID Becas/Doctorado Nacional 21212258 scholarship and ES was funded by a scholarship from Vicerrectoría de Investigación de la Escuela de Graduados, Pontificia Universidad Católica de Chile.

## Author contributions

Conceptualization, X.T., R.P.M., M.J.A., R.A.M., and G.A.; Methodology, X.T., R.P.M., and G.A; Validation, X.T.; Formal Analysis, X.T., R.P.M., G.A., H.L.B., and T.M.C.; Investigation, X.T., R.P.M., G.A., M.J.A., and R.A.M.; Cohort study design, supervised and managed the sample collection, E.F.S., T.G.S., C.P.R., A.R. and R.A.M.; Processed samples, revised the manuscript, E.F.S., T.G.S., C.P.R., J.L., E.P., E.S., A.M., A.R., and R.A.M.; Resources, G.A. and R.A.M.; Writing—Original Draft, X.T. and R.P.M.;

Writing—Review & Editing, R.P.M., G.A., M.J.A., and R.A.M.; Visualization, X.T., R.P.M., and G.A.; Project Administration, R.P.M.; Funding Acquisition, G.A. and R.A.M.; Supervision, R.P.M., R.A.M., and G.A.

## Competing interests

The authors declare the following competing interests; Galit Alter is a founder/equity holder in Seroymx Systems and Leyden Labs. G.A. has served as a scientific advisor for Sanofi Vaccines. G.A. has collaborative agreements with GSK, Merck, Abbvie, Sanofi, Medicago, BioNtech, Moderna, BMS, Novavax, SK Biosciences, Gilead, and Sanaria. R.A.M. has served as a scientific advisor for Valneva SE. The remaining authors declare no competing interests.
