## [Peer Review File · Nature Communications]

Waning and boosting of antibody Fc-effector functions upon SARS-CoV-2 vaccinationREVIEWER COMMENTS

Reviewer #1 (Remarks to the Author):

Tong et al provide a good comparison of two major types of vaccines in terms of antibody levels over (relatively short period of) time (5M), as well as their antibody reactions to the low affinity FcγR and their potential to activate complement. Reaction/cross reactivities to the major VOC is also investigated. The vaccine comparison and crossreactiveness form the most informative part of this study. The rationale of looking into the reactivity to the different low affinity FcγR is completely not given. In fact, the binding activity of these receptors is simply given when in real life the biological activities of these receptors (except that of FcγRIIIa on NK cells) always occurs in concert with other co-expressed receptors on myeloid cells. The only rationale known why it would be interesting to look the binding activities of these receptors, is because of altered glycosylation (almost exclusively fucosylation) these vaccine may promote. In that case only including FcγRIIIa (and perhaps FcγRIIa, or simply total IgG as surrogate for quantity), is interesting. However, even though the data seem to suggest the FcγR reactivities goes more or less hand in hand with IgG levels and sensitivity issues governed by the affinity of the different FcγR to IgG, this is not mentioned nor compared. The relative sensitivity of FcγRIIb for IgG4 is not mentioned, and neither is the sensitivity of FcγRIIIa/b for afucosylated IgG, although some part of the data presented may suggest this has some relevance.

Major points

-p7: "These data suggest that mRNA boosting can broaden the subclass, isotype, and FcR binding profile across VOCs (Supplementary Figure 2), but may only partially rescue FcR binding specifically to more distant VOCs." Your data are in line with a large body of literature that antibodies raised other original vaccine strains are capable of (sometimes limited) generating cross-reactive antibodies, some of which are blocking. This is naturally always less than to the original strain. Boosting certainly helps to overcome this, partially by increased affinities of the resulting antibodies. It is inconceivable that that the capacity of these antibodies has less FcγR activities, but because you have less, you see less in your FcγR readout. The question becomes how sensitive your assay is to quantity of antibodies and relative to FcγR readout. This is not shown or discussed.

-It is also remarkable that you mention nothing about the rationale using FcγR individually a readout (see general remark and major point above) especially with regard to IgG-afucosylation which is the principal factor affecting binding to FcγRIIIa/b and with prepublications suggesting that some vaccination (mRNA) do induces afucosylated responses to a variable degree. This fits with what was suggested by two groups last year in Science and Nat Immunol paper, showing that afucosylated responses in SARS-CoV-2 infection do occur and matter. This becomes almost a necessity to look at this in light of figure 4 for the CoronaVac boost looking at both WT and Omicron NK cell activity, that gets an immense boost. That would fit an afucosylated response, no other explanation known in the field would fit the bill better.

Minor points

-abstract: when referring to levels found in CoronaVac comparing to those in BNT162b2 in the waning period of the former, indicate which time point you compare to for the latter.

-introduction, when referring to protection rate, I think you should consider the difference in vaccine strain vs the major circulating strain at the time of study. These are likely to be different and influence these numbers.

-referring to any particular of these receptors as cytotoxic receptors e.g. top page 6) is a misconception (except excluding this for the inhibitory FcγRIIb and the GPI-linked FcγRIIIb), as all the other FcγR are potentially capable mediated cytotoxic responses, it just depends on the context (which receptors are expressed on which cells, e.g. NK cells can't mediate phagocytosis). The same goes for "opsonophagocytic" which applies to all except FcγRIIb.

-referring to antibodies specific for the inhibitory receptor, or FcγRIIb (top page 6), these do not exist. All antibodies (IgG) are specific for their antigen, with IgG1 and IgG3 binding all FcγR, IgG2 mostly FcγRIIa. Thus your formulation is incorrect. You probably don't see binding in your assays as these two receptors are the two most low affinity receptors (for IgG1, the major isotype you are likely looking at).

-When referring to the FcγR proteins as you do, you should not use alphanumeric designation,

but roman numerals, e.g. FcγRIIB, not FcγR2B (γ standing for gamma naturally).-Supplementary figures sometimes have [γ] instead of a [gamma]

-sup fig 2A mentions columns but the figures does not contain columns.

- Why do you leave FcγRI out of the picture?

-The relative increased binding to FcγRIIb is heralded as robust, but if this is increased this would suggest IgG4 response is increased (fitting with your data) which does not bode well for strong effector functions.

-Similarly increased relative FcγRIIIa binding suggests most likely afucosylation responses, but this should also correlate with IIIb readout. Did you compare this? Especially interesting seeing that the crossreactive antibodies have relatively higher IIIa profile

-Fig 4 seem unnecessary complicated as why do you have gray/blue , gray/purple if they stand for WT/omicron spike in both cases??? Please also include the actual individual data.

Reviewer #2 (Remarks to the Author):

General Comments

1. The authors present a study of SARS CoV2 systems serology profiling Fc effector profiles of an inactivated, adjuvanted whole virus vaccine, and mRNA vaccine. The kinetics of antibody were measured, and this included post whole virus boosting using mRNA vaccine. Side effect profiles were not provided for both vaccines, which is an omission given the vast differences between the vaccine formulations.

2. Some of the logic expressed in the paper is unclear, for example in the first paragraph, the remarkable success of vaccines has little to do with virus adaptations, except in a teleological sense. The manuscript needs to be reviewed for logical expression.

3. The authors do not consider other vaccine candidates in their discussion, which should be broadened.

Specific Comments

- The authors need to clearly distinguish efficacy (reduced disease in studies) from effectiveness (real world performance). They use both terms, although they have very different implications.
 - P3 Vaccine numbers are freely available, and "half a dozen" is inappropriate
 - P3 If vaccines differ in the nature of protective efficacy induced, can the mRNA boost be truly said to "restore the waned" whole virus immunity?
 - P5 The serological responses are age dependent (eg Selva KJ Nature Comms 2021) and although unlikely to be significant, the age differences should be clear – the booster group ages are not provided on p14 methods.
 - P5 "FcγR2B, FcγR3A, and FcγR3B-binding Spike- specific levels were induced more slowly after the first mRNA-vaccine dose, requiring the second dose to mature fully". Is this necessarily dependent, or is it a time-dependent association? That is, is there a requirement/dependence or not? What are the data that support this?
 - P6 Why was there a similar waning in RBD specific antibodies with mRNA and whole virus vaccines? This seems surprising given the different nature of the two vaccines. Is it possibly due to consistency in RBD sequences found over the pandemic?
 - P9 Can the authors describe the "uniqueness" of the opsinophagocytic activity raised by mRNA vaccines here?
 - P10 It is incorrect to state CoronaVac provides a "unique" priming strategy. There are several other whole virus vaccines (Covi-Vac, VLA2001) including killed (BBV152) and killed adjuvanted vaccines such as COVIran Berekat.
-
- Figure 1 Were technical replicates duplicate or triplicate? Latter should be later.
 - The significant variability of individual responses is a difficult confounder in such studies.

Reviewer #3 (Remarks to the Author):

The manuscript by Tong et al, is an overall comprehensive analysis of the antibody binding (Ig isotypes, FcγR-2A,-2B,-3A, and -3B) response to SARS-CoV-2 detected by Luminex and FcR-mediated functional responses (ADCP, ADNP, ADCD, and ADNKA; by Flow) in a Chilean cohort of COVID-19 vaccinees. In specific, these parameters are longitudinally analysed in two groups: one group (n=15) vaccinated with BNT162b2 vaccine (Pfizer) as per protocol 2 administrations 21 days apart, and the other group (N=34) vaccinated 30-days apart 2 times with Coronavac vaccine, of which a subgroup (N=20) received a booster with BNT162b2.

The results show that Coronavac vaccination without booster is substantially inferior to BNT162b2 vaccination in terms of peak and waning of binding to Wild type (WT) and VOCs SARS-CoV-2 spike, FcγR to WT, and functional responses to WT and Omicron. The breadth of the functional responses (defined with WT and VOC Omicron) is similar for both vaccines, except for the phagocytic activities ADNP and ADCP mediated by THP1 or neutrophils, which are better for BNT162b2. While the booster to Coronavac improves considerably all responses, also towards the VOC Omicron. No comparison is done with a boost after the BNT162b2 2x vaccine regimen. The Authors conclude that a vaccination regimen Coronavac - BNT162b2 may be relevant for those situations when only Coronavac was used.

The techniques are well established in the laboratory by Galit Alter, and previously published in the context of SARS-CoV-2 infection and vaccination.

This is a first report on functional FcγR mediated responses in Coronavac vaccinees cohorts, while those in BNT162b2 vaccinees was elsewhere described. The study of the booster response is relevant to drive further vaccination protocols. This Chilean cohort was object of other studies, in specific for binding and neutralizing antibody and T cell responses. Thus, the Authors may want to comment in the discussion the other papers on this cohort, specially with regard to the relevance of the neutralizing antibody response.

Major Comments:

The cohort and in specific the sampling of serum samples needs a more detailed description. These data may affect the statistical analysis and in addition add some more information to the paper, though they may not alter the overall message.

The cohort: No mention is done if study participants had COVID-19 previous to vaccination or between vaccination and booster, and if equally distributed in the two groups of vaccinees. As already previously published by Alter and others, a previous SARS-CoV-2 infection may affect the rapidity and peak of the humoral subsequent responses.

It would be more informative if the baseline data were not pooled in one single baseline group. This data set could also be separated in the baseline box of all figures.

It would be relevant to mention the time-interval between vaccination and Booster. Though only 20 subjects, it may show some difference in the response according to time.

Sample collection time-points: Specify vaccination timepoints; for example, when were the "1- and 2-dose BNT162b2 mRNA" samples collected? How many days after vaccination? same for 1- and 2-dose Corevac.

The authors state that the follow-up samples were collected between 1 and 209 days after first or second vaccination for BNT162b2 or Corevac, respectively; however, in the results/figures timepoints are analysed at 2-3 and 4-5 months, which suggests that a time-interval of 2 months was used for statistical analysis. A more precise definition of timing and selection of the samples for analyses and statistics, should be provided.

Analogously, it is never mentioned if the same number of longitudinal samples were included for each participant. Individual data points in the boxes of the figures show a higher number than that of the number of participants for each group. Please specify. Means and statistical analysis would be affected by uneven samples and length of follow-up for each participant.

Data of Figure 4 may be expanded and detailed in a supplementary figure, in which individual data points are included, to show the variability for each timepoint of the functional responses. The variability should be commented in the results section and eventually discussed.

Supplement Figure 1: shows antibody binding responses to other pathogens, such as seasonal betacoronavirus, Influenza and Ebola. These results are briefly mentioned in the result section. It would be relevant to analyse and discuss if differences of these responses (specially to OC43 or HKU1) in the two groups of vaccinees may drive responses to the two vaccine-regime, as elsewhere published for other vaccine regimen.

Methods:

For the functional assays it is not specified if all WT spike antigens listed in Supplementary Table 1 (i.e. Spike, S1 and S2 domain) were tested, but results shown only for the Spike. Table should be corrected if only the spike was used, or data shown if all antigens were used.

Figures:

Supplementary figure 3: if space allows, the panels of Supp Fig 3 should be added to Figure 3. The results of these two figures are in line and consistent.

Figure 4, panels B and C: the Y-axis legends of the two panels are identical = "phagocytic score". Though correct, for ease of reading it would be preferable to add ADNP and ADCP in B and C Y-axis legend, respectively.

In general, figures are clear, but figure legends could be simplified, when the same text is used for each panel.

Point-by-point response to the reviewers' comments

Reviewer #1 (Remarks to the Author):

Tong et al provide a good comparison of two major types of vaccines in terms of antibody levels over (a relatively short period of) time (5M), as well as their antibody reactions to the low-affinity FcγR and their potential to activate complement. Reaction/cross-reactivities to the major VOC are also investigated. The vaccine comparison and cross-reactiveness form the most informative part of this study. **The rationale for looking into the reactivity to the different low-affinity FcγR is completely not given.** In fact, the binding activity of these receptors is simply given when in real life the biological activities of these receptors (except that of FcγRIIIa on NK cells) always occur in concert with other co-expressed receptors on myeloid cells. The only rational known why it would be interesting to look at the binding activities of these receptors is because of **altered glycosylation** (almost exclusively **fuco-sylation**) these vaccines may promote. In that case, only including FcγRIIIa (and perhaps FcγRIIa, or simply total IgG as a surrogate for quantity), is interesting. However, even though the data seem to suggest the FcγR reactivities go more or less hand in hand with IgG levels and **sensitivity issues** governed by the affinity of the different FcγR to IgG, this is not mentioned nor compared. The relative sensitivity of **FcγRIIB for IgG4** is not mentioned, and neither is the sensitivity of FcγRIIIa/b for afucosylated IgG, although some parts of the data presented may suggest this has some relevance.

Major points

Comment 1

p7: “These data suggest that mRNA boosting can broaden the subclass, isotype, and FcR binding profile across VOCs (Supplementary Figure 2) but may only partially rescue FcR binding specifically to more distant VOCs.” Your data are in line with a large body of literature that antibodies raised by other original vaccine strains are capable of (sometimes limited) generating cross-reactive antibodies, some of which are blocking. This is naturally always less than to the original strain. Boosting certainly helps to overcome this, partially by increased affinities of the resulting antibodies. It is inconceivable that the capacity of these antibodies has fewer FcγR activities, but because you have less, you see less in your FcγR readout. The question becomes how sensitive your assay is to the quantity of antibodies and relative to FcγR readout. This is not shown or discussed.

Responses:

We thank the reviewer for this comment concerning the sensitivity and robustness of our assays to quantify the antigen-specific antibodies and binding to the FcγRs. It is a valid critique of the reviewer considering that we are measuring multiple antigen subclasses and FcγRs at the same time. Regarding the general sensitivity and robustness of the assays used in this study, our systems serology pipeline is Good Clinical Laboratory Practice (GCLP) aligned and undergoes annual validation (PMID: 35881018, 34824251, 35868417). To further address this comment, we have also added a new statement in our **Methods** section regarding the general sensitivity of our assays used in this study.

Page 23, Line 561-567, “The general sensitivity and robustness of the assays used in this study to quantify the antibody titers and relative FcγR binding capacity and specifically in the context of SARS-CoV-2 infections and vaccines have been previously validated (PMID: 24927273,

35348368, 36000735, 35881018). The sensitivity of the assays used specifically to evaluate the antibody isotypes and the receptor binding was also validated by the dilution curves generated prior to the analysis of the samples, as shown in **Supplementary Figure 1.**”

In addition, we have also added a new **Supplementary Figure 1** to provide the dilution curves used to determine the linear ranges of the serum dilution factors for the antibody and FcγR binding measurements.

“**Supplementary Figure 1. Dilution curves for the antibody isotyping and receptor binding capacity.** The linear ranges of the serum dilution factors were determined for IgG1, IgM, IgA, and FcγRIIIA (3AV). Samples from each subgroup were randomly selected and assayed against the Spike protein of the Wild-type, Alpha, Beta, Gamma, Delta, and Omicron variants. All samples were assayed in technical duplicates.” (**Supporting Information, Figure S1**).

Comment 2

It is also remarkable that you mention nothing about the rationale using FcγR individually as a readout (see general remark and the major point above), especially with regard to IgG-afucosylation which is the principal factor affecting binding to FcγRIIIa/b and with prepublications suggesting that some vaccinations (mRNA) do induce afucosylated responses to a variable degree. This fits with what was suggested by two groups last year in *Science* and *Nat Immunol* paper, showing that afucosylated responses in SARS-CoV-2 infection do occur and matter. This becomes almost a necessity to look at this in light of figure 4 for the CoronaVac boost looking at both WT and Omicron NK cell activity, which gets an immense boost. That would fit an afucosylated response, no other explanation known in the field would fit the bill better.

Responses:

We thank the reviewer for this comment and apologize for the confusion on our part. We thank the reviewer for providing us with a potential explanation for the mechanism by which binding to FcγR could affect the NK cell responses against both the WT and Omicron Spike induced by the mRNA booster following CoronaVac vaccination. As suggested, the factor of IgG-afucosylation would fit our observations in the functional responses shown in Figure 4. Therefore, we have performed the glycan analysis on samples from the naïve, peak immunity, the lowest decayed, and heterologous boosted subgroups. The results of glycan analysis are highly consistent with the reviewer’s proposed mechanism and are summarized in the newly incorporated **Supplementary Figure 5**. The following paragraphs of glycan analysis were added to the **Results, Discussion, and Methods** sections.

Results, Pages 14-15, Lines 308-338, “Given the emerging importance of antibody glycosylation as an important factor affecting the functional activities of vaccine-induced antibody responses against COVID-19 (39, 40), we next aimed to investigate the specific mechanism(s) by which the NK activity-enhancing antibodies was selectively induced in the boosted samples compared to the other groups. We conducted glycan analysis on samples from the naïve, at peak immunity, at the lowest decayed, and for the heterologous boosted subgroups (**Supplementary Figure 5**). The relative abundances of *N*-glycan modifications were characterized, including the lack of fucose (afucosylation), bisecting *N*-acetyl glucosamine

(GlcNAc), and galactose. Notably, anti-WT Spike IgG1 from individuals who received the first regimen of two doses of CoronaVac or BNT162b2 was reduced in core fucosylation when compared with the naïve group. Furthermore, those who received the third dose of heterologous boosting exhibited a further reduction in the level of fucosylation after the waning period, in agreement with previous studies (41). We also validated the results of reduced fucosylation by analyzing the afucosylation levels of subjects receiving the BNT162b2 vaccine, which revealed similar patterns. In addition, we also observed significant increases in bisecting glycans and galactosylation following vaccination and heterologous boosting. Collectively these results are consistent with previous studies analyzing the relationship between the antibody glycosylation profiles and immune responses against SARS-CoV-2 infections and vaccinations.”

Discussion Page 18, Lines 398-426, “Previous studies have demonstrated that the levels of IgG afucosylation, galactosylation, and bisecting GlcNAc differed in cohorts of COVID-19 when meta-analysis were performed (PMID: 33361116, PMID: 33169014). The level of bisecting GlcNAc was the most prominent feature distinguishing severe and mild COVID-19. Decreased bisection was found in severe patients. Higher levels of bisecting GlcNAc on IgG were reported to indirectly affect affinity for FcγRs and enhance ADCC by inhibiting fucosylation. Substantial changes in antibody galactosylation levels in severe COVID-19 infection result in a higher abundance of galactosylated IgG molecules compared to mild COVID-19, and such changes are related to the pro-inflammatory effects of IgG. This proinflammatory function of galactosylation on the one hand acts through complement system activation. Previous studies reported decreased galactosylation both in anti-S IgG1 and total IgG1 in severe COVID-19 compared to mild, which is consistent with changes in the total IgG glycome (PMID: 36069511). Collectively, our results of glycan analysis are therefore consistent with a wide range of previous studies focusing on IgG glycome and COVID-19 infection.”

Methods, Page 26, Lines 630-648, “**Glycosylation analysis,** Antigen-specific glycosylation analysis was performed as described previously (PMID: 35402869). Briefly, the WT Spike antigen was biotinylated and coupled to NeutrAvidin magnetic beads at a ratio of 2.5 μg of protein to 25 μL of beads. Combined serum (200 μL) was first incubated with non-coated NeutrAvidin beads to remove the non-specific binding and then added to the antigen-coated beads and incubated for 1 h at 37°C. Antibodies were eluted by incubation in 50 μL of pH 2.9 citrate buffer for 30 min at 37°C. Samples were then spun down and the eluted antibodies, contained in the supernatant, were neutralized with 30 μL pH 8.9 potassium phosphate buffer. Antibodies were then coupled to protein G beads. After the beads were washed, IDEZ was used to cleave the Fab (in the supernatant) from the Fc (remained on the magnetic beads) for 1 h at 37°C and collected. The two fragments were deglycosylated and fluorescently labeled using a GlycanAssure APTS kit according to the manufacturer’s instructions. For the Fc fragment, which remained bound to the protein G beads, an additional magnetic separation after PNGase glycan cleavage separated the glycans from the remaining magnetic beads and the protocol then proceeded per the manufacturer’s instructions. Glycans were analyzed on a 3500xL genetic analyzer. Glycan fucosyl and afucosyl libraries were used to assign 24 discrete glycan peaks using GlycanAssure software. Data are reported as percentages of total glycans for each of the glycan peaks.”

Minor points
Minor Comment 1

Abstract: when referring to levels found in CoronaVac compared to those in BNT162b2 in the waning period of the former, indicate which time point you compared to for the latter.

Responses:

We thank the reviewer for pointing out this ambiguity of our statement. We have modified the sentences in the Abstract to be more precise regarding the time points that were compared between the two subgroups.

Abstract, Page 2, Lines 30-32, “Here, using systems serology, we profiled the Fc-effector profiles induced by the CoronaVac and BNT162b2 vaccines up to five months following the two-dose vaccine regimen”.

Minor Comment 2

Introduction: when referring to protection rate, I think you should consider the difference in vaccine strain vs the major circulating strain at the time of the study. These are likely to be different and influence these numbers.

Responses:

We thank the reviewer for this comment. We agree of the important to consider the difference in vaccine strain vs the major circulating strain at the time of the study. To address this comment, we have added a statement to the Introduction to clarify the different protection rates of vaccines against VOCs.

Introduction, Page 4, Line 82-94, “Both clinical trials were conducted in late 2020 and early 2021 in which period Alpha and Gamma were the circulating VOCs in South America, where the study was conducted. Literature reviews and meta-analyses suggested that full vaccination with CoronaVac or BNT162b2 provided strong protection against the Alpha, Beta, Gamma, and Delta variants with a vaccine effectiveness ranging from 70.9% to 96.0% against severe disease. The two vaccines also provided moderate protection against the recently emerged Omicron variant” (PMC8279092).

Minor Comment 3

Referring to any particular of these receptors as cytotoxic receptors e.g., **top page 6**) is a misconception (except excluding this for the inhibitory FcγRIIb and the GPI-linked FcγRIIIb), as all the other FcγR are potentially capable mediated cytotoxic responses, it just depends on the context (which receptors are expressed on which cells, e.g., NK cells can't mediate phagocytosis). The same goes for “opsonophagocytic” which applies to all except FcγRIIb.

Responses:

We thank the reviewer for this comment and for pointing out the misconception about the FcγR functions. We have reworded our language in the Results section and removed “cytotoxic” and “opsonophagocytic” where appropriate throughout the manuscript.

Minor Comment 4

Referring to antibodies specific for the inhibitory receptor, or FcγRIIb (top page 6), these do not exist. All antibodies (IgG) are specific for their antigen, with IgG1 and IgG3 binding all FcγR, IgG2 mostly FcγRIIa. Thus, your formulation is incorrect. You probably won't see binding in

your assays as these two receptors are the two most low-affinity receptors (for IgG1, the major isotype you are likely looking at).

Response:

We thank the reviewer for this comment. It is a valid statement by the reviewer to point out that we cannot conclude in our formulation which IgG subclasses are responsible for binding to the majority of the low-affinity Fc γ Rs. Therefore, to clarify this point, we have added a new statement to the Discussion as part of the limitation with this study.

Discussion, Page 19, Lines 454-456, “It is noteworthy that the measurements shown here for the IgG subclassing and receptor binding were independently performed. Therefore, the antibody Fc-mediated binding activities are not assigned to an IgG subclass.”

Minor Comment 5

When referring to the Fc γ R proteins as you do, you should not use alphanumerical designation, but roman numerals, e.g. Fc γ RIIB, not Fc γ R2B (g standing for gamma naturally). Supplementary figures sometimes have [y] instead of a [gamma]

Response:

We thank the reviewer for this comment and for pointing out this mistake in the designation. Therefore, we have corrected the designation of Fc γ Rs throughout the manuscript. We have also corrected all Fc γ Rs in the Supplementary Information.

Minor Comment 6

Supplementary fig 2A mentions columns but the figures do not contain columns.

Response:

We thank the reviewer for this comment and for pointing out this mistake in the legends of Figures and Supplementary Figures regarding “columns”. We have replaced all the “columns” in the Figures with “Lanes” to be more accurate in describing the components of the Figures.

Comment 7, Why do you leave Fc γ RI out of the picture?

Response:

We appreciate this comment by the reviewer and agree that Fc γ RI could be potentially interesting to study as the Fc γ RI expression is associated with interferon levels. However, as part of the limitation of our platform, the binding to high-affinity receptors often depends on the titer and it's not as specific as the low-affinity Fc γ Rs used in our platform. Therefore, we have added a statement in the Discussion section to propose a future study to investigate the relationship between Fc γ RI and antibody functional responses.

Discussion, Page 19, Lines 456-459, “In addition, future studies on the relationship between the high-affinity Fc γ RI and antibody functional responses could potentially further clarify the overall humoral responses towards COVID-19”.

Comment 8-9,

- the relative increased binding to Fc γ RIIb is heralded as robust, but if this is increased this would suggest IgG4 response is increased (fitting with your data) which does not bode well for strong effector functions.

- Similarly increased relative FcγRIIIa binding suggests most likely afucosylation responses, but this should also correlate with IIIb readout. Did you compare this? Especially interesting to see that the crossreactive antibodies have a relatively higher IIIa profile.

Response:

We appreciate these two related comments by the reviewer. First, we agree that increased binding to FcγRIIb could potentially attenuate the effector functions. We acknowledge that most likely multiple antibody subclasses interacting with a group of different FcγRs are contributing to the overall effector functional responses. As we explained in the response to Minor Comment 4, above, we cannot conclude from our formulation which IgG subclasses are responsible for binding to the majority of low-affinity FcγRs. This is also true for the relationship between FcγRIIIa/b and afucosylation responses. As mentioned in the response to Major Comment 2, above, we have performed glycan analysis on the samples, and the results are consistent with Reviewer's proposed mechanism.

Comment 10

Fig 4 seems unnecessarily complicated why do you have gray/blue, gray/purple if they stand for WT/omicron spike in both cases??? Please also include the actual individual data.

Response:

We thank the Reviewer for this valuable comment. We agree that **Figure 4** could be further simplified in terms of color labels. We have thus modified the **Figure 4** to be more simplified and included actual data points instead of the original trend representation.

Reviewer #2 (Remarks to the Author):

General Comments

General Comment 1

The authors present a study of SARS CoV2 systems serology profiling Fc effector profiles of an inactivated, adjuvanted whole virus vaccine, and mRNA vaccine. The kinetics of antibodies were measured, and this included post-whole virus boosting using an mRNA vaccine. Side effect profiles were not provided for both vaccines, which is an omission given the vast differences between the vaccine formulations.

Responses:

We appreciate this general comment by the reviewer. We thank the reviewer for pointing out the omission about the side effects of these two vaccines. Of note, our study was not a vaccine clinical trial, hence the side effects of vaccination was not recorded for this cohort, instead we collected the demographic data of the study participants which is provided in Supplementary Table 2. To address this comment, we have added a paragraph regarding the side effect profiles of these two vaccines in the **Introduction** section.

Introduction, Page 5, Lines 94-100,“According to data published from their respective clinical trials, the most common adverse effects following COVID-19 vaccination include fatigue, fever, muscle pain, headache, and joint pain. More serious side effects were rarely

recorded and reported. The clinical trial of the CoronaVac inactivated vaccine reported frequent injection site pain followed by fever and other mild and self-limiting conditions. The Pfizer-BioNTech mRNA vaccine trial has reported that the most common adverse effect was mild to moderate fatigue and headache (PMID: 35659687).”

General Comment 2

Some of the logic expressed in the paper is unclear, for example in the first paragraph, the remarkable success of vaccines has little to do with virus adaptations, except in a teleological sense. The manuscript needs to be reviewed for logical expression.

Responses:

We appreciate this general comment by the reviewer. We have checked the manuscript for logical expression and made correction accordingly. Specific to this comment, we have reworded the sentence in the first paragraph as below:

Introduction, Page 3, Lines 45-47, “However, despite the remarkable success of vaccines in protecting the population from the early emergent SARS-CoV2 viral strains, the virus has undergone adaptations that have facilitated transmission among humans”.

General Comment 3

The authors do not consider other vaccine candidates in their discussion, which should be broadened.

Responses:

We thank the reviewer for this comment and for pointing out our limitation of not including other vaccine candidates in our discussion. Therefore, we have added a new paragraph to the **Discussion** section regarding the heterologous boosting studies involving other vaccine candidates.

Discussion, Page 17, Lines 385-397, “In addition to the CoronaVac and Pfizer/BNT162b2 vaccines, previous studies have also suggested that heterologous boosting between other COVID-19 vaccines could potentially enhance the humoral immune responses against the disease (PMID: 35081293, PMID: 34370971, PMID: 35072129). A vaccine boosting study demonstrated that seven different COVID-19 vaccines, including ChAdOx1 (Oxford-AstraZeneca), NVX-CoV2373 (Novavax), BNT162b2 (Pfizer–BioNTech), VLA2001 (Valneva), Ad26.COV2 (Janssen), mRNA1273 (Moderna), and CVnCoV (CureVac) are safe and induce strong immune responses when administered as booster doses following two doses of either BNT162b2 or ChAdOx1 vaccines. Another study compared the safety and immunogenicity of a third heterologous booster dose of either the ChAdOx1, BNT162b2, or Ad26.COV2 vaccines in adults in Brazil who previously received two doses of CoronaVac. These results collectively indicated that a robust anamnestic immune response can be induced by each of these vaccines when used as a booster regardless of the primary COVID-19 vaccination regimen (PMID: 35074136).”

Specific comments

Specific Comment 1

The authors need to clearly distinguish efficacy (a reduced disease in studies) from effectiveness (real-world performance). They use both terms, although they have very different implications.

Response:

We thank the reviewer for this comment and for pointing out differences between efficacy and vaccine effectiveness in our manuscript. We agree that these two terms have very different implications in describing the features of vaccine candidates. Therefore, we have revised the uses of these two terms in the manuscript and have corrected the terms where they were used inaccurately throughout the text.

Specific Comment 2

P3 Vaccine numbers are freely available, and “half a dozen” is inappropriate

Response

We thank the reviewer for this comment. We agree that “half a dozen” is inappropriate in describing the vaccine numbers. Therefore, we have removed “half a dozen” and added a more accurate description of vaccine numbers to the **Introduction** section.

Introduction, Page 3, Lines 41-45, “Since it was first identified in late 2019 (1-3), 11 COVID-19-specific vaccines, using novel and diverse platforms, have been granted the emergency use listing (EUL) by WHO to provide protection against this highly transmissible pathogen, and four of these COVID-19 vaccines are currently approved or authorized in the United States (4).”

Specific Comment 3

P3 If vaccines differ in the nature of protective efficacy induced, can the mRNA boost be truly said to “restore the waned” whole virus immunity?

Response

We appreciate this comment from the Reviewer. We agree that it might not be appropriate to state that the mRNA “restored the waned” whole virus immunity if these two vaccines differ in the nature of their protective mechanisms. Therefore, we have changed the statement to a new one shown below:

Introduction, Page 5, Line 106-111, “Here we deeply profiled the functional humoral immune response induced by CoronaVac and Pfizer/BNT162b2 vaccines. Particularly we analyzed how the vaccine-induced functional responses waned with time and characterized the boosting capacity of the BNT162b2 vaccine, which enhanced the overall humoral responses of CoronaVac primed vaccinees to levels that were higher than the peak responses seen in the recipients vaccinated with two doses of the CoronaVac or BNT162b2 vaccines.

Results, Page 12, Line 269, “mRNA-Vaccine boosting of CoronaVac recipients broadens functional humoral defenses”

Page 13, Line 295-298, “Again, while low Omicron ADCP was induced by the CoronaVac vaccine that waned, a cross-reactive response was induced with the BNT162b2 immunization, which is comparable to the WT-specific primary ADCP levels induced by the boost.”

Discussion, Page 15, Lines 355-357, “Yet critically, a Pfizer/BioNTech boost of CoronaVac immunized recipients expanded the antibody effector function, in some cases above those observed with mRNA immunization alone.”

Specific Comment 4

P5 The serological responses are age-dependent (e.g. Selva KJ Nature Comms 2021) and although unlikely to be significant, the age differences should be clear – the booster group ages are not provided on p14 methods.

Response

We appreciate this comment from the Reviewer. We agree that age differences could be important for the discussion here. Therefore, to address this comment, we have added a new **Supplementary Table 2** to show the demographics of the cohort. As can be seen, the age and sex of the individuals are very similar in both groups (median age for each group and the overall cohort was between 33-36 years, and females corresponded to 70-80% of the groups).

Specific Comment 5

P5 “FcγR2B, FcγR3A, and FcγR3B-binding Spike- specific levels were induced more slowly after the first mRNA-vaccine dose, requiring the second dose to mature fully”. Is this necessarily dependent, or is it a time-dependent association? That is, is there a requirement/dependence or not? What are the data that support this?

Response

We appreciate this comment from the Reviewer. We agree that the statement might not be accurate in determining the dependence of time. Therefore, we have modified the sentence in the Results section to a more accurate expression.

Results, Page 8, Lines 174-175, “Conversely, FcγRIIB, FcγRIIIA, and FcγRIIIB-binding Spike-specific levels exhibited peak responses after the second dose of vaccine (**Figure 2B-D**).”

Specific Comment 6

P6 Why was there a similar waning in RBD-specific antibodies with mRNA and whole virus vaccines? This seems surprising given the different nature of the two vaccines. Is it possibly due to consistency in RBD sequences found over the pandemic?

Response

We appreciate this comment from the Reviewer. We agree that a similar waning in RBD-specific antibodies with mRNA and whole virus vaccines is surprising given the different immunological induction elicited by these two vaccines. We have added a new paragraph in the **Discussion** to address this comment.

Discussion, Page 20, Lines 469-479, “Studies have shown that the neutralizing activity of vaccine-elicited antibodies could be focused on the receptor-binding domain (RBD) as compared to infection-elicited antibodies. However, within the RBD, the binding of vaccine-elicited antibodies is more broadly distributed across epitopes than those seen for infection-elicited antibodies. The rapid decay of RBD-specific antibodies could be due to the fact that RBD is a smaller sub-domain of the Spike protein. Therefore, targeted waning could be more

rapid for RBD than for the full-length Spike, independent of the vaccine platform. This greater binding breadth suggests that single RBD mutations have less impact on neutralization by vaccine sera than convalescent sera. Hence, antibody immunity acquired by different means may have differing susceptibility to erosion by viral evolution (PMC8057239, 32841599, 34192529).”

Specific Comment 7

P9 Can the authors describe the “uniqueness” of the opsinophagocytic activity raised by mRNA vaccines here?

Response

We appreciate this comment from the Reviewer. For clarity we have removed the word “uniqueness” from this sentence used in the **Discussion** section.

Specific Comment 8

P10 It is incorrect to state CoronaVac provides a “unique” priming strategy. There are several other whole virus vaccines (Covi-Vac, VLA2001) including killed (BBV152) and killed adjuvanted vaccines such as COVIran Berekat.

Response

We thank the Reviewer for this comment. We have modified this sentence in the Discussion section to include the other whole virus vaccines. Please also refer to General Comment 3, Reviewer 2 above for additional information regarding the other whole virus vaccines involved in heterologous boosting studies.

“Moreover, similar to other vaccines based on whole viral particles, this vaccine represents a priming strategy, given the exposure to all viral components, many of which may provide additional protection in the setting of future viral variation.” (**Discussion**)

Specific Comment 9

Figure 1 Were technical replicates duplicate or triplicate? Latter should be later.

Response

We thank the Reviewer for this comment. All samples were assayed in technical duplicates for Luminex assays. For functional assays all samples were assayed in technical quadruplicates with a minimum of three independent donors. We have added these two statements throughout the manuscript. We have changed the word “latter” to “later”.

Specific Comment 10

The significant variability of individual responses is a difficult confounder in such studies.

Response

We thank the Reviewer for this comment. We agree that the significant variability of individual responses is a difficult confounder in these types of studies. We have added a statement to the Discussion section to address this comment.

Discussion, Page 20, Line 479-483, “In addition, the significant variability of individual responses is a considered a difficult confounder in these types of studies. We deliberately grouped samples together to account for heterogeneity between individual responses. Therefore,

statistical grouping based on this method should be considered as conservative approach given the known discrepancy in these responses.”

Reviewer #3 (Remarks to the Author):

General Comments

The manuscript by Tong et al, is an overall comprehensive analysis of the antibody binding (Ig isotypes, FcyR-2A,-2B,-3Av, and -3B) response to SARS-CoV-2 detected by Luminex and FcR-mediated functional responses (ADCP, ADNP, ADCD, and ADNKA; by Flow) in a Chilean cohort of COVID-19 vaccinees. In specific, these parameters are longitudinally analysed in two groups: one group (n=15) vaccinated with BNT162b2 vaccine (Pfizer) as per protocol 2 administrations 21 days apart, and the other group (N=34) vaccinated 30-days apart 2 times with Coronavac vaccine, of which a subgroup (N=20) received a booster with BNT162b2.

The results show that Coronavac vaccination without booster is substantially inferior to BNT162b2 vaccination in terms of peak and waning of binding to Wild type (WT) and VOCs SARS-CoV-2 spike, FcyR to WT, and functional responses to WT and Omicron. The breadth of the functional responses (defined with WT and VOC Omicron) is similar for both vaccines, except for the phagocytic activities ADNP and ADCP mediated by THP1 or neutrophils, which are better for BNT162b2. While the booster to Coronavac improves considerably all responses, also towards the VOC Omicron. No comparison is done with a boost after the BNT162b2 2x vaccine regimen.

The Authors conclude that a vaccination regimen Coronavac - BNT162b2 may be relevant for those situations when only Coronavac was used.

The techniques are well established in the laboratory by Galit Alter, and previously published in the context of SARS-CoV-2 infection and vaccination.

This is a first report on functional FcyR mediated responses in Coronavac vaccinees cohorts, while those in BNT162b2 vaccinees was elsewhere described. The study of the booster response is relevant to drive further vaccination protocols. This Chilean cohort was object of other studies, in specific for binding and neutralizing antibody and T cell responses. Thus, the Authors may want to comment in the discussion the other papers on this cohort, specially with regard to the relevance of the neutralizing antibody response.

Major Comments:

Comment 1. The cohort and in specific the sampling of serum samples needs a more detailed description. These data may affect the statistical analysis and in addition add some more information to the paper, though they may not alter the overall message.

Response: We thank the reviewer for this comment, which helps clarify the study design and cohort. To address this, we have included two additional **Supplementary Tables 2 and 3** describing the grouping of the samples and their demographics and detailing the sampling times for each individual included in the study. We have also included additional information in the methods section to describe the cohort and the serum samples used in more detail.

Methods, Page 21, Lines 523-534, “The cohort contains samples from individuals who received either BNT162b2 (n = 15) or CoronaVac vaccines (n = 34) (Supplementary Table 2 and 3). The BNT162b2 mRNA vaccine group was given 30 µg BNT162b2 (15 – 53 years old, median: 36 years, 80% female) on days 0 and 21, and serum samples were taken up to 168 days after the second dose. The CoronaVac group (21 – 80 years old, median: 33 years old, 70.6% female) received two doses of 600 U CoronaVac four weeks apart, and individuals were sampled up to 209 days after the second dose. A subgroup of CoronaVac subjects (n = 23, 23 – 80 years old, median: 35 years old, 65.2% female) received a booster dose of the BNT162b2 mRNA vaccine and was sampled 14-31 days after the mRNA booster. We did not observe any immunocompromising comorbidities associated with the cohort.”

Comment 2, the cohort: No mention is done if study participants had COVID-19 previous to vaccination or between vaccination and booster, and if equally distributed in the two groups of vaccinees. As already previously published by Alter and others, a previous SARS-CoV-2 infection may affect the rapidity and peak of the humoral subsequent responses.

Response:

We appreciate this comment, as this is an important point to clarify. All the individuals included in the study were naïve at the time of vaccination, hence previous exposure to SARS-CoV-2 was ruled out by a qRT-PCR test at the time of recruitment. Only 3 individuals reported to have COVID-19 between vaccine dose 1 and 2 (1 individual) or after the second dose (2 individuals). This information was added to the **Methods** section. We have also added a new paragraph to address the possible exposure to SARS-CoV-2 prior, before, and during this study.

Methods, Page 21, Lines 535-546, “All the individuals included in the study were naïve at the time of vaccination, hence previous exposure to SARS-CoV-2 was ruled out by a qRT-PCR test at the time of recruitment. Only 3 individuals reported to be exposed to COVID-19 between vaccine dose 1 and 2 (1 individual) or after the second dose (2 individuals). In addition, we monitored for infection-acquired anti-nucleocapsid antibodies throughout the study. Since CoronaVac contains nucleocapsid, a noticeable and expected response was observed exclusively within this group. Moreover, nucleocapsid antibodies waned progressively over five months within this group, arguing against any exposure to SARS-CoV2 during this period. No nucleocapsid responses were observed in the BNT162b2 arm as this mRNA vaccine only encodes for Spike protein. We thus collectively conclude that SARS-CoV2 infections prior, before, and during this study were minimal-to-absent. (**Supplementary Figure 3A**).”

Comment 3. It would be more informative if the baseline data were not pooled in one single baseline group. This data set could also be separated in the baseline box of all figures.

Response:

We appreciate this comment by the reviewer. To address this, in **Supplementary Table 2** we now show the demographics of the cohort. As can be seen, the age and sex of the individuals are very similar in both groups (median age for each group and the overall cohort was between 33-36 years, and females corresponded to 70-80% of the groups). We also separated the baseline data (the pre vaccine data point) and tested these baseline data separately for both groups for any statistical difference among the groups (n=19 naïve sera for CoronaVac vaccinees and n=14 for

the BNT162b2 vaccinees). We found no statistical differences in the background data (see the newly modified **Supplementary Figure 3**). Given that all these individuals were confirmed to be negative at the time of recruitment and had no previous history of COVID-19, and they all correspond to one cohort sampled concurrently; we propose that these data should be used as a single pooled baseline, which is representative of the entire cohort and also allows greater statistical power for the subsequent analyses.

Comment 4. It would be relevant to mention the time-interval between vaccination and Booster. Though only 20 subjects, it may show some difference in the response according to time.

Response:

We thank the reviewer for this suggestion. We have added a new **Supplementary Table 3**, where we included the exact days and time intervals of the initial vaccination and the booster for all the participants in the study.

Comment 4. Sample collection time-points: Specify vaccination timepoints; for example, when were the “1- and 2-dose BNT162b2 mRNA” samples collected? How many days after vaccination? same for 1- and 2-dose Corevac.

Response:

We thank the reviewer for this comment. As stated in Comment 3, we have now included these data on a newly incorporated **Supplementary Table 2**, which includes specific data for each individual. Briefly, for the BNT162b2 vaccinees the “dose 1” samples were collected between days 20-27 post first dose and before the second dose and the “dose 2” samples were collected between days 13-19 after second dose. For the CoronaVac vaccinees the “dose 1” samples were collected between days 27-30 post first dose and before the second dose, and the “dose 2” samples were collected between days 13-31 after second dose. To complement **Supplementary Table 2 and 3. The Figure 1A** was re-made to represent this information better.

Comment 5. The authors state that the follow-up samples were collected between 1 and 209 days after first or second vaccination for BNT162b2 or Coronavac, respectively; however, in the results/figures time points are analysed at 2-3 and 4-5 months, which suggests that a time-interval of 2 months was used for statistical analysis. A more precise definition of timing and selection of the samples for analyses and statistics, should be provided.

Response:

We thank the reviewer for this comment. We agree with the reviewer that this was not totally clear. In our study design, there were two time points considered for the long-term follow up, a 3-month and a 6-month time points. For the 3 months’ time point the samples were collected between days 59 – 104 post second dose, hence these were considered the 2–3-month results analyzed in the manuscript. For the 6 months’ time point the samples were collected between days 115 – 168 post second dose. Hence, these were considered the 4–5-month results analyzed and shown in the figures. We have also incorporated this information in the newly incorporated **Supplementary Figure 3** and is also shown on the new **Figure 1A**.

Comment 6. Analogously, it is never mentioned if the same number of longitudinal samples were included for each participant. Individual data points in the boxes of the figures show a higher number than that of the number of participants for each group. Please specify. Means and statistical analysis would be affected by uneven samples and length of follow-up for each participant.

Response:

We thank the reviewer for this comment. We agree with these comments, and hence we have now provided additional data for each of the groupings we used for the analyses. In addition, we have added the new **Supplementary Table 2 and 3** including the number of individuals per group, demographic characteristics of the cohort and detail the specific time intervals (days) of samples obtained from vaccinated individuals.

Comment 7. Data of Figure 4 may be expanded and detailed in a supplementary figure, in which individual data points are included, to show the variability for each timepoint of the functional responses. The variability should be commented in the results section and eventually discussed.

Response:

We thank the reviewer for this comment. Reviewer #2 also raised a similar issue. To address this comment, we have re-made **Figure 4**, which exhibits individual data points. The variability for each time point of the functional responses is included in the newly incorporated **Figure 4** and the modified Figure legend as well as in the Results and Methods sections.

Comment 8. Supplement Figure 2: shows antibody binding responses to other pathogens, such as seasonal betacoronavirus, Influenza and Ebola. These results are briefly mentioned in the result section. It would be relevant to analyze and discuss if differences of these responses (specially to OC43 or HKU1) in the two groups of vaccinees may drive responses to the two vaccine-regime, as elsewhere published for other vaccine regimen.

Response:

We thank the reviewer for this comment. To further address the raised topic regarding the antibody binding responses to other pathogens, we have performed new statistical analysis on the results shown in the modified **Supplementary Figure 3**. While minor changes in OC43 reactive antibodies were noted in **Supplementary Figure 3**. No such changes reached statistical significance through the multiple test correction. A higher number of individuals enrolled in this study could potentially change our interpretation of the current data. However, at this point we believe our results demonstrated that the functional responses shown in this study were attributable to the SARS-CoV2-directed antibodies.

Comment 9. Methods:

For the functional assays it is not specified if all WT spike antigens listed in Supplementary Table 1 (i.e. Spike, S1 and S2 domain) were tested, but results shown only for the Spike. Table

should be corrected if only the spike was used, or data shown if all antigens were used.

Response:

We thank the reviewer for this suggestion. **Supplementary Table 1** was corrected, and we ensured that all materials are listed correctly as used in this study.

Comment 10. Figures:

Supplementary figure 3: if space allows, the panels of Supp Fig 3 should be added to Figure 3. The results of these two figures are in line and consistent.

Response:

We thank the reviewer for this suggestion. We have thus added panels of the previous **Supplementary Figure 3** to the newly incorporated **Figure 4**. The results of this new Figure 4 are now consistent as suggested by the Reviewer.

Comment 10

Figure 4, panels B and C: the Y-axis legends of the two panels are identical = “phagocytic score”. Though correct, for ease of reading it would be preferable to add ADNP and ADCP in B and C Y-axis legend, respectively.

Response:

We thank the reviewer for pointing this out. “ADNP” and “ADCP” were added to the “phagocytic score” in the Y-axis legends of panels B and C of the modified **Figure 4** as suggested by the Reviewer.

Comment 11

In general, figures are clear, but figure legends could be simplified when the same text is used for each panel.

Response:

We thank the reviewer for this comment. We have simplified the figure legends and ensured that the same text is not repetitively used in the figure legends

REVIEWERS' COMMENTS

Reviewer #1 (Remarks to the Author):

I appreciate the hard work the authors have put into this also supplying new data. I still have some concerns. I note that the level of reported glycosylation is way off in comparison with other studies (much lower), perhaps due to methodological differences. Can the authors comment on this why this is, and how this can be skewed? Also the referred studies do not make sense (line 289, "Furthermore, those who received the third dose of heterologous boosting exhibited a further reduction in the level of fucosylation after the waning period, in agreement with previous studies (41)). Ref 41 originates from last century way before anyone had ever achieved antigen-specific analysis of IgG glycosylation, and certainly way before the introduction of any mRNA vaccine. Please correct.

The text now in lines 353-358 is littered with inaccuracies. "Previous studies have demonstrated that levels of IgG afucosylation, galactosylation, and bisecting GlcNAc differed in cohorts of COVID-19 when meta analysis was performed (39, 40). The level of bisecting GlcNAc was the most prominent feature distinguishing severe and mild COVID-19. Decreased bisection was found in severe patients. Higher levels of bisecting GlcNAc on IgG were reported to indirectly affect affinity for FcγRs and enhance ADCC by inhibiting fucosylation." First of all – neither of these studies performed a true meta analysis. Second the level of bisecting glcnac was not the most prominent feature distinguishing the two groups (severe and mild). There were some significant differences, but these were small, and are functionally irrelevant as reported by multiple studies. In these studies, afucosylation was the most prominent difference early in the response (first week of seroconversion, not later) in which time only the severe patients showed a dramatic decrease in galactosylation. Third the effect of bisecting glcnac inhibiting fucosylation has only been reported in one CHO cell line, not in HEK and not in human responses.

Reviewer #2 (Remarks to the Author):

In reviewing the response of the authors to Reviewer #3:
The authors response to Reviewer #3

General Comments – Did not respond to the comment "the Authors may want to comment in the discussion the other papers on this cohort, specially with regard to the relevance of the neutralizing antibody response"

This was not done, but is not critical

Comment 1 – Adequate inclusion in methods and 2 supplementary table

Comment 2 – Adequate, cohort appropriate

Comment 3 – Adequate, did not change figures but added demographics appropriate

Comment 4 – Adequate

Comment 5 – Adequate, although would have been helpful to add to the text

Comment 6 – Adequate, again a sentence to include "the same number of longitudinal samples were included for each participant. " would be helpful

Comment 7 – Adequate, additional data included in Figure 4

Comment 8 – Adequate, they have not added text to the paper, but have added a supplementary figure, and my view is this is reasonable, although not necessary

Comment 9 – Adequate

Comment 10 – Adequate

Comment 11 – Adequate

Comment 1

I appreciate the hard work the authors have put into this also supplying new data. I still have some concerns. I note that the level of reported glycosylation is way off in comparison with other studies (much lower), perhaps due to methodological differences. Can the authors comment on this why this is, and how this can be skewed?

Response:

We thank the reviewer for this comment concerning the levels of antibody glycosylation reported in this study. It is a valid critique of the reviewer considering that the glycosylation levels of polyclonal antibodies are often reported by different groups using various analytical techniques. Traditionally, high-performance liquid chromatography and mass spectrometry-based techniques have been used for N-glycan analysis of antibodies. Our study applied a capillary electrophoresis-based technique to analyze the antibody glycosylation profiles of antibodies. One possible reason behind the difference could be that previously described analytical techniques antibody glycan analysis have focused on the analysis of whole IgG, whereas our methodology allows for separate analysis of the N-glycans from Fc and Fab domains. Therefore, our reported glycan levels could be lower from other studies that are reporting the glycan levels of whole antibodies. In addition, the reported values of antibody glycans in this study are within a similar range of several previously published studies concerning humoral signatures of antibodies during SARS-CoV-2 infections and vaccinations in humans using the same capillary electrophoresis-based analytical technique (PMID: 33589825, PMID: 36781879, and PMID: 36351430).

Comment 2

Also the referred studies do not make sense (line 289, “Furthermore, those who received the third dose of heterologous boosting exhibited a further reduction in the level of fucosylation after the waning period, in agreement with previous studies (41)). Ref 41 originates from last century way before anyone had ever achieved antigen-specific analysis of IgG glycosylation, and certainly way before the introduction of any mRNA vaccine. Please correct.

Response:

We thank the reviewer for this comment for pointing out this mistake. We have corrected the mistake in the reference studies. The Reference 41 should be PMID: 36529104. This recently published study demonstrated that mRNA vaccines induced an increase in the afucosylated IgG response after the first and second dose of the BNT162b2 mRNA, which agrees with our results shown here.

Comment 2

The text now in lines 353-358 is littered with inaccuracies. “Previous studies have demonstrated that levels of IgG afucosylation, galactosylation, and bisecting GlcNAc differed in cohorts of COVID-19 when meta-analysis was performed (39, 40). The level of bisecting GlcNAc was the most prominent feature distinguishing severe and mild COVID-19. Decreased bisection was found in severe patients. Higher levels of bisecting GlcNAc on IgG were reported to indirectly affect affinity for FcγRs and enhance ADCC by inhibiting fucosylation.” First of all, neither of these studies performed a true meta-analysis. Second the level of bisecting GlcNAc was not the most prominent feature distinguishing the two groups (severe and mild). There were some significant differences, but these were small, and are functionally irrelevant as reported by multiple studies. In these studies, afucosylation was the most prominent difference early in the response (first week of seroconversion, not later) in which time only the severe patients showed a dramatic decrease in galactosylation. Third the effect of bisecting GlcNAc inhibiting fucosylation has only been reported in one CHO cell line, not in HEK and not in human responses.

Response:

We thank the reviewer for this comment for this comment. We apologize for the inaccurate points in this statement. We have replaced the original inaccurate text in lines 353-358 with the following sentences.

“Previous studies have demonstrated that relative levels of afucosylation, galactosylation, and bisecting GlcNAc of antibodies differ in response to SARS-CoV-2 vaccination or infection (PMID: 34883043, PMID: 33169014, PMID: 33979301, and PMID: 3336111). The recent studies collectively suggest that the level of the afucosylated glycan represents the most prominent feature in association with the disease severity in the early response. Subjects with severe COVID-19 symptoms exhibited a higher proportion of afucosylated spike-specific antibodies than patients with mild disease. The antibody fucosylation levels then waned 4-6 weeks following symptom onset (PMID: 34883043, PMID: 33169014, PMID: 33979301, and PMID: 3336111). Levels of both, antibody galactosylation and sialylation were also found to be upregulated in the infected individuals and displayed correlation with disease severity (PMID: 3336111, PMID: 35334306). Following infection or vaccination, levels of spike-specific bisecting GlcNAc remained the same or showed transient decrease, and decreased levels of bisecting GlcNAc were found to be associated with recovery from previous infection (PMID: 34883043, PMID: 33169014, PMID: 3336111, and PMID: 35334306). Collectively, our results of glycan analysis are therefore consistent with a wide range of previous studies focusing on IgG glycome and COVID-19 infection.”

Reviewer #3

General Comments – Did not respond to the comment “the Authors may want to comment in the discussion the other papers on this cohort, especially with regard to the relevance of the neutralizing antibody response”.

Response:

We thank the reviewer for this comment. To address this comment, we added a sentence in the Discussion section describing the neutralizing antibody response we determined in our previous study, *EBioMedicine*. 78:103972. DOI: 10.1016/j.ebiom.2022.103972.

Discussion, Page 16, Lines 323-327, “We have previously shown that the two-dose vaccination scheme with CoronaVac induces neutralizing antibody titers similar to a single mRNA dose immunization, but significantly lower than those of a two-dose mRNA vaccination. Here we show that this extends to the Fc effector function...”